# Computer Self-Efficacy and Reactions to Feedback: Reopening the Debate in an Interpretive Experiment with Overconfident Students [note 1]

**DOI:** 10.3390/bs15040511

**Published:** 2025-04-11

**Authors:** Carlo G. Porto-Bellini, Malu Lacet Serpa, Rita de Cássia de Faria Pereira

**Affiliations:** 1Department of Business Administration, Universidade Federal da Paraíba, João Pessoa 58051-900, Brazil; rita.pereira@academico.ufpb.br; 2Human Resources Management, Universidade Federal de Pernambuco, Recife 50670-901, Brazil; malu.serpa@ufpe.br

**Keywords:** computer self-efficacy, overconfidence, technology use effectiveness, task performance, feedback, learning, experiment, interpretive research, analytical reasoning

## Abstract

The accuracy of self-perceptions and the maturity to handle feedback received from others are instrumental for one’s mental health, interpersonal relations, and effectiveness in the classroom and at work. Nonetheless, research on one’s computer self-efficacy (CSE) and reactions to feedback on task performance has been ambiguous in terms of quality, motivations, and results. A particularly important case involves overconfident individuals, i.e., those with unrealistically high CSE beliefs. Using the theoretical lenses of technology use effectiveness along with a mixed methodological approach of thought and lab experiments with 54 undergraduate students performing computer-aided tasks who were randomly assigned to different groups receiving feedback on task performance, we found that valence-based feedback possibly introduces unnecessary information to adjust the levels of CSE and the actual performance of overconfident students. This finding contributes to knowledge on whether feedback is important when skills and learning naturally mature across tasks, in addition to how judicious one is when processing externally motivated feedback. This study additionally offers a novel three-dimensional CSE construct, an instrument to measure the construct, and a scenario-based tool to conduct experiments with sequential decision tasks in the classroom. The practical implications include the planning of tasks and feedback in the classroom, with further insights into organizational hiring, training, and team building.

## 1. Introduction

Self-perceptions and external perceptions about the self are natural concerns for all individuals, since perceptions are closely related to self-confidence and self-esteem for the emergence of substantive adulthood, mental health, and social life ([20]). Moreover, developing appropriate self-perceptions may help avoid the emergence of extreme attitudes, including suicidal ideation, as reported in all parts of the world and across cultures (e.g., [96]; [68]; [66]; [55]; [76]). In the field of human–computer interaction, an important part of perceptions about the self involves how individuals see their own capabilities to use digital technologies to achieve arbitrary goals. According to the seminal models on technology use, individuals will use a system based on self-efficacy beliefs and perception-based attitudes towards that system and towards the use environment (e.g., [25]; [21]; [26]; [93]). Such a framing of technology use is more motivational in nature and ignores whether the user is effectively capable of operating the system in reference to stated purposes, i.e., whether the user has the needed theoretical abilities and practical skills in addition to certain positive attitudes towards the system and the use environment.

We designed a study on how technology users perceive themselves and react to external perceptions about themselves (from feedback provided by an authorized agent) regarding their actual performance in computer-aided tasks. We aimed to estimate how mindful one is about performing tasks on the computer and how he or she processes others’ assessments about his or her performance and correspondingly adjusts attitudes and behaviors. In doing so, we contribute to the emergent stream of research on technology use effectiveness ([15]; [71]), in some sense improving the dominant models of technology use by framing not only what motivates use but also what makes one achieve arbitrarily defined use purposes by understanding what needs to be achieved and the means to do so.

Moreover, we have paid special attention to the presence of overconfidence in individuals performing computer-aided tasks, which is a particularly important trait in the study of human cognition and decision-making ([45]). Overconfidence is here framed as a post hoc measure of unrealistically high self-efficacy beliefs, which represents a cognitive limitation towards one’s effectiveness in using digital technologies. We therefore searched for an answer to the question of whether feedback on performance, as well as the valence of feedback, has any impact on overconfident individuals in their computer self-efficacy (CSE) beliefs and actual task performance. To answer it, we designed an experiment with undergraduate students of Business Administration who were regularly assigned to perform computer-aided decision tasks. We submitted the participants to two sequential, similar computer-aided decision tasks and measured three newly defined dimensions of CSE before each task (general CSE, problem-specific CSE, and tool-specific CSE) and, after each task, we measured their self-perceptions of performance as well as their actual performance. In between the two tasks, we provided each participant with one of three types of manipulated feedback: neutral, negative, and positive feedback on their task performance. An important characteristic of the experiment is that it involved a realistic computer-aided business decision under real assessment by an authorized agent (the course’s instructor), instead of any computer use. With such an experimental design, we studied technology use through the theoretical lenses of use effectiveness, which requires the definition of the purposes of use and assumes that more than one perspective of use effectiveness may exist (in the present study, the perspective of each student and the perspective of their instructor).

This study contributes to the literature on many fronts. First, it empirically investigates the effects of feedback received from supervisors (instructors, managers, or team leaders) regarding one’s performance in computer-aided problem solving as well as by comparing the levels of actual performance with the levels of CSE. It is also among the few studies, if any, discussing the role of (motivational) attitudes, (theoretical) abilities, and (practical) skills. It additionally gives answers to calls for research on providing feedback on digital skills ([69]; [1]), on learning issues being “hindered by biased attention allocation and overconfidence” ([30]), on the effects of immediate feedback over student learning ([19]), on the relation of one’s mindset, reactions to feedback, learning, and performance ([23]; [22]), and on the relation of self-efficacy, feedback, and self-regulation ([47]). Moreover, it contributes to the long-lived debate about the relationship between confidence and accuracy ([36]), particularly whether CSE predicts technology use effectiveness ([73]). For practice, this study has numerous implications for students, instructors, and organizational managers, such as regarding the assignment of computer-aided tasks to students and workers and the appraisal of their performance.

This article is organized as follows. In the literature review section, we present the key constructs of this study, i.e., CSE, overconfidence (a special case of CSE), task performance, feedback, and learning. Next, we discuss a mixed experimental design to study how feedback received from supervisors may impact an individual’s CSE, self-perceptions on task performance, and actual performance. In developing the methodological approach, we propose that CSE can be measured in three dimensions to better isolate and understand specific issues in computer-aided decision-making. Subsequently, we analyze and discuss the empirical data according to the approach of analytical reasoning, which is characterized by insightful, ad hoc procedures to process unusual datasets. And finally, we discuss the implications of our findings as well as limitations and perspectives for future research.

## 2. Literature Review

A review by [93] ([93]) concerning the first decade of research on the unified theory of acceptance and use of technology (UTAUT) showed that studies on information technology (IT) acceptance, adoption, and use relied on perceptions about a system’s features, the use environment, or someone’s self-efficacy. However, perceptions do not necessarily correspond to reality, much like how in-use technologies may not correspond to espoused ones ([67]). In fact, people may hold unrealistic views about technology, the use environment, and even about themselves, such as when expected performance is not confirmed vis-à-vis actual performance ([9]). We thus resort to the works of [15] ([15]) and [71] ([71]) on technology use effectiveness to discuss the presence of individuals in educational and professional settings who may not hold accurate views about themselves in terms of the actual competencies needed to perform a task on the computer. The use effectiveness perspective extends traditional views of technology use by focusing on how mindful one is about his or her technology-related motivational attitudes, theoretical abilities, and practical skills. Moreover, effectiveness is a relativistic measure of the level in which someone achieves his or her arbitrarily defined technology use purposes, i.e., each stakeholder in a given technology use situation will hold a particular view of effectiveness in the light of his or her personal perspectives about what effectiveness is (what he or she wishes to do) in that particular situation ([71]).

In the next sections, we review the seminal works and research opportunities on the key concepts of our study (self-efficacy beliefs, task performance, feedback, and learning) and connect them with the technology use effectiveness rationale. Self-efficacy beliefs are instrumental for one to achieve effectiveness in technology use (an individually defined expectation about task performance). Feedback, in turn, is an opportunity to demonstrate that different perspectives may exist about task performance and that the stakeholders must negotiate a shared expectation as per the demands of the situation. For instance, students and instructors may hold different views about acceptable student performance in the classroom. While all views are acceptable as per the concept of technology use effectiveness ([71]), the stakeholders must share a common expectation if a larger stakeholder is present (in the case of classroom activities, the larger stakeholder is the school’s evaluation system). Finally, learning is a natural process through which people adapt their views about the world and about themselves in order to meet those shared expectations.

### 2.1. Computer Self-Efficacy

Self-efficacy refers to one’s judgment about the competencies needed to perform tasks in view of expected performance ([7]). Relatedly, the construct of computer self-efficacy (CSE) refers to the ability/efficacy to use a computer to complete a task ([21]; [57]). While CSE assessments do not reveal one’s actual skills, the levels of CSE may be related to one’s motivation to develop the needed skills to use a computer in effective ways ([87]), thus impacting how much effort, persistence, and interest the individual will invest in a task, as well as the type of environment in which the individual desires to work ([34]).

An important issue is how to measure CSE. To the best of our knowledge, [64] ([64]) and [35] ([35]) have proposed the first operational models of CSE. [64] ([64]) conceived CSE in three factors, namely *beginning level computer skills*, *advanced level computer skills*, and *mainframe computer skills*. Such factors address general and specific computer skills as well as skills related to the technological apparatus in use. [35] ([35]) followed a similar rationale to conceive CSE in two factors, namely *computer self-efficacy* (general skills about the computer context) and *software self-efficacy* (skills about a specific tool). [21] ([21]) then discussed CSE within the interests of the IT literature and defined it parsimoniously as a single-factor, 10-item construct addressing task issues. Soon after, [57] ([57]) elaborated CSE as an IT concept conveying two constructs (*general CSE* and *task-specific CSE*) that “cannot be treated interchangeably from either a measurement or manipulation perspective” (p. 129). A decade later, [28] ([28]) criticized all available CSE measurement models and developed a task-based, *summated general CSE*. However, their model was specific to six arbitrarily chosen tools and conceived the tasks as the mere operation of those tools, i.e., they did not address for which purposes the tools were used―the real tasks of interest. Subsequently, in a meta-analysis of 102 empirical CSE studies, [46] ([46]) concluded that serious problems existed in how CSE had been defined and measured, but they did not propose any alternative model. [38] ([38]) revisited [57]’s ([57]) CSE model as it had not given full attention to the tasks that are performed with the mediation of technology. They thus proposed that CSE should be measured according to the *specificity of technology* and the *complexity of the task*. More recently, the hypotheses that guided [21]’ ([21]) seminal work on CSE were submitted to a replication study three decades after the original study and with the participation of one of the original authors ([85]). While the original rationale for the construct was preserved, about half of the original hypotheses were not accepted in the replication. And finally, [88] ([88]) developed a CSE scale based on a framework by the European Commission to address multiple heterogeneity problems found in their literature review. Their newly developed CSE construct (called *digital self-efficacy*) includes the following five dimensions: information and data literacy, communication and collaboration, digital content creation, safety, and problem solving. It is apparent to us, however, that two of these dimensions may not be necessary in stand-alone IT uses or when a tool is used by one individual only, i.e., not by interacting individuals. These dimensions are safety and communication and collaboration.

It is thus apparent that CSE is still a construct undergoing conceptual refinement and empirical validation. We advocate that, in the light of technology use effectiveness, the concept of CSE should merge the rationale of all previous models so as to include three distinct components based on the interplay of reality and representation ([15]), as follows: (1) the real-world purpose of technology use, (2) the computer as the representational context of that purpose, and (3) the specific software tool as the representational mechanism to aid the user in achieving the stated purpose. We will offer a way to implement these three components when describing the experimental design of our empirical study. We will call the three components, respectively, *problem-specific* CSE, *general* CSE, and *tool-specific* CSE.

### 2.2. Self-Efficacy and Performance

Even if a positive correlation was originally expected to exist between self-efficacy beliefs and actual task performance ([6]), self-regulation theories were criticized in the past for being silent on certain self-efficacy levels contributing to individuals allocating less resources than needed to successfully complete a task ([91], [92]). Indeed, debate still exists on whether self-efficacy is positively related to performance in all situations or if the negative effects occasionally found are merely the visible side of an adaptive process of judicious and efficient allocation of finite resources ([9]). It seems, though, that high levels of self-efficacy promote excessively optimistic feelings about performance, which in turn results in unrealistic perceptions that one is near to goal achievement ([80]) or far from risks for the self and for others ([61]), or even that future performance can repeat successful past performance ([40]).

A special situation occurs when self-efficacy beliefs exceed a certain level and make an individual report overconfidence in performing a task. Overconfidence may reduce performance of the task ([62]). This is particularly true for individuals who have been successful in previous decisions but, in certain moments, pay less attention to available cues and more attention to personally held information, thus incurring in increased risks of lower decision quality ([40]). As a result, overconfidence has been said to account for the erroneous self-assessment of skills ([30]). Overconfidence is generally seen as the (1) overestimation of one’s actual performance, (2) overplacement of one’s performance relative to the performance of others, or (3) overprecision in one’s beliefs ([61]). An alternative view of overconfidence is a motivational one, whereby individuals “are often more confident than accurate (…) a desire to see the self as knowledgeable and competent” ([12]), meaning that “people typically overestimate their prowess” ([30]) and that their “confidence is typically higher than is justified by the observed accuracy” ([36]). This may be one reason why studies on the relationship between self-efficacy beliefs and actual performance do not find positive correlations under certain conditions (e.g., [70]; [80]; [90]; [91], [92]; [97]). As such, self-efficacy is conceptually challenging, and, for the interests of the present study, the particular case of overconfidence is framed as a cognitive limitation for one’s technology use effectiveness ([71]), i.e., an unrealistically high estimation of one’s actual abilities to perform computer-aided tasks.

### 2.3. Feedback and Learning

A related concern is feedback. Providing feedback on abilities and actual performance is believed to help regulate perceptions and behaviors, promote learning and progress ([54]; [83]; [4]; [36]), and shape one’s intrinsic motivation to perform ([77]). In general learning situations, feedback has been used to confirm and reinforce the correct paths of action and adjust the incorrect ones ([83]). In organizations, feedback is an integral aspect of work and an old interest of scholars ([54]), whereby peers and supervisors provide information on one’s success or failure in performing tasks either in regard to task fulfilment itself or to the individual’s particular performance ([89]), with impacts on competence validation and competence development ([59]). In schools, feedback is also an important part of the learning process ([65]) and of educational performance ([24]), with students who receive feedback from instructors and engage in self-regulated learning practices performing better ([86]). Nevertheless, feedback was also found to inhibit learning in some situations ([60]).

However, it is not any kind of feedback that will possibly promote learning and progress. The literature is broadly inconclusive as to the existence of a dominant pattern of the impact of feedback on human performance ([84]). In one of the most cited reviews on feedback, [39] ([39]) concluded that both the type of feedback and the way it is given play specific roles in learning and achievement. The authors summarize feedback in three questions (“going where”/feed up, “going how”/feed back, and “what next”/feed forward) and four categories (about either the task, the processing of the task, self-regulation, or the self), and they state that it is at the level of self-regulation feedback that confidence and self-efficacy beliefs mediate feedback reception and effectiveness. They also address how cultural issues may impact feedback design and outcomes, as well as the need to appropriately plan the timing, the valence, the optimal use, and the role of assessment in feedback interventions. Such planning is needed to increase one’s effort, motivation, and engagement, as well as to minimize the negative impacts of feedback on self-efficacy and performance if proper cause–performance paths are not communicated to those receiving feedback.

Discussions on the complexity and maturity of the feedback literature are not new. [50] ([50]) developed feedback intervention theory motivated by about 90 years of poor scholarly knowledge on how feedback relates to behavior and performance, as well as by evidence that feedback significantly reduces performance in certain situations or does not have any effect at all—whereas the widely accepted assumption is that feedback improves performance. To those authors, feedback interventions are “actions taken by (an) external agent(s) to provide information regarding some aspect(s) of one’s task performance” (p. 255), but related research for almost a century was plagued with inconsistencies regarding definitions, methodology, analyses, and the resulting admissible inferences. Also, they found that many experimental studies on feedback lacked control groups, thus not allowing for a comparison between the effects of feedback with no feedback at all. [50] ([50]) affirmed that, by that time, there was no solid knowledge on how feedback (any feedback or none, its type, signal, magnitude, or focus) acted on individuals to adjust the perceptions about the self and task performance. Further problems included issues like (1) researchers ignoring high variances in the findings so as to highlight only the positive outcomes of feedback; (2) the use of varying standards to design and assess feedback; (3) researchers ignoring the locus of attention across feedback interventions, such as whether feedback was directed to the process, to the outcomes, or to the agents performing the tasks; and (4) the simplistic assumption that any behavior could be regulated by feedback. Moreover, there are so many idiosyncratic situations in the relationship between feedback, people, tasks, and their environment that it would be impossible with current knowledge and methods to predict specific outcomes from feedback. Finally, those authors identified the lack of a task taxonomy, thus developing one based on the following variables: subjective novelty, intrinsic complexity (number of actions and dependencies), time constraint, needed creativity, performance measurement (quality or quantity, objective data or subjective ratings), transfer (if the effect is measured on a subsequent task), latency (latency/speed), and type of task (physical, reaction, memory, knowledge, rule-based, vigilance).

The current literature remains hesitant on several aspects of feedback, such as its affective components and outcomes. For instance, encouraging/positive feedback has been reported to help students achieve the generally desirable state of flow, but it may also create counterproductive situations ([16]). In turn, negative feedback has been reported to be negatively associated with learning ([24]) and to increase the levels of stress and negative affect while having no significant impact on one’s performance or self-esteem ([84]). Mixed findings also exist regarding the impacts of feedback frequency ([53]) and feedback timing ([5]) on learning and task performance. Additional design issues of feedback interventions include the following: absolute/individualized or relative/comparative feedback ([10]; [99]), whether feedback is directed to an individual or to a group ([75]), feedback clarity ([13]; [79]; [11]), feedback intensity ([33]), feedback consistency ([56]), the perceived credibility of the source of feedback ([94]), the degree of motivation of those who provide feedback ([79]), whether feedback comes from peers ([100]), automated devices ([41]) or a combination of human and automated processes ([84]), and if feedback is immediate/interactive ([47]) or used in massive learning environments ([69]).

Worthy of note, the outcomes of feedback also depend on the individuals who receive it, as they need to engage in reflection by having a learning goal orientation, motivation to think extensively, and a feeling of task-performance importance ([4]) along with self-efficacy beliefs significantly related to feedback-seeking behaviors ([82]; [27]). Also influential are their self-concepts ([58]) and whether they hold a fixed/stable/entity or an incremental/malleable/growth view of abilities/intelligence ([29]). Furthermore, evidence reported by [48] ([48]) shows that an individual’s cognitive resources predict feedback requirements, in the sense that more capable individuals (in task demands) would benefit more from reduced feedback, whereas less capable individuals would benefit more from increased feedback.

Lastly, feedback may be contrasted with feedforward and whether an individual passively receives or actively seeks feedback[note 1]. Feedforward refers to one’s intentions for the future rather than to performance in the past ([14]), while feedback seeking refers to a change in the locus of control in the feedback process ([3]). In our study, none of the issues is present, as we focus on assessing past performance (feeding back) and on feedback that is designed and motivated by the researchers’ interests, i.e., those who provide, not seek, feedback.

On the methodological front, the use of experiments to study the effects of feedback on overconfidence in computer settings is not new (e.g., [98]; [65]), but a recent review by [78] ([78]) identifies a paucity of experimental studies on the relationship between social stressors and performance in human–computer interaction, with negative feedback on performance being among those stressors. Therefore, one of our contributions to the literature is to study, in an experimental design, the effects of valence-based feedback on actual performance in computer-aided tasks as well as on three newly defined domains of CSE to verify if overconfident individuals regulate appraisals of self-efficacy and become more aware of the needed resources to fulfil a task. In doing this, we can inform instructors on how to help students reflect on their learning performance from the early stages of educational development, and managers to design worker–task fit and training more judiciously.

## 3. Method

We designed an experiment to study the effects of a supervisor’s feedback on a subordinate’s performance in computer-aided tasks, with a particular interest in the reactions of overconfident individuals. We modelled the tasks at a subjectively intermediate level of complexity due to the ambiguous relationship between task complexity and one’s self-appraisals: on one hand, easy tasks are said to preserve overconfidence ([30]) and make people underestimate actual performance while also believing that they are better than others ([61]); on the other hand, difficult tasks are said to promote greater self-insight ([30]) and make people overestimate actual performance while also believing that they are worse than others ([61]). Therefore, we did not anticipate the signal and the magnitude of the possible impact of feedback on the levels of CSE and actual task performance of overconfident individuals. For tasks with an intermediate level of complexity, our hypotheses are as follows:

**H1a.** 
*Overconfident individuals adjust their CSE beliefs after receiving feedback on performance from a supervisor.*


**H1b.** 
*Overconfident individuals adjust their self-reported performance after receiving feedback on performance from a supervisor.*


**H1c.** 
*Actual task performance of overconfident individuals is impacted after they receive feedback on performance from a supervisor.*


Additional hypotheses may be stated for the effects of skills and learning that naturally mature across tasks, as follows:

**H2a.** 
*Overconfident individuals adjust their CSE beliefs as per their practical skills and learning across similar tasks.*


**H2b.** 
*Overconfident individuals adjust their self-reported performance as per their practical skills and learning across similar tasks.*


**H2c.** 
*Actual task performance of overconfident individuals is impacted as per their practical skills and learning across similar tasks.*


The experimental design included features of thought and lab experiments. Lab experiments are popular in IT research, but thought experiments are less so, being a form of qualitative research ([44]) for theory development ([37]). Our experiment includes both designs due to the presence of testable (H1a, H1b, H1c) and conjectured (H2a, H2b, H2c) causal paths. The lab experiment part has features of a field experiment due to the experimental groups being studied in their natural setting, i.e., during a real execution of computer-aided tasks and with no information provided to the performing individuals about the experimental activities. We informed the participants about the experiment only after its conclusion, when we asked for their individual consent to analyze and publish the data[note 2]. The lab experiment part was also a true experiment ([17]) due to the randomization process and other features described next.

Participants were undergraduate students of Business Administration assigned to two different classrooms (morning and evening sections) of a course called Administrative Informatics. A total of 54 students participated in this study, 29 from the morning section and 25 from the evening section. We assigned them evenly and randomly to three groups: 18 students to an experimental group that would receive positive feedback on task performance (G_POS_), 18 students to an experimental group that would receive negative feedback (G_NEG_), and 18 students to a control group that would receive neutral feedback (G_CTRL_). In addition to there being an equal number of students in each group, their assignment to the groups was partially controlled for pairing, i.e., the demographic profile of each group was partially homogenized regarding the most typical variance-generating variables as judged from the historical demographic distribution in similar classrooms (Table 1). Full pairing of all variables was evidently not an option, since pairing the most influential variables impedes full pairing of other, less influential ones. For instance, one may question why gender was not a priority for pairing. The reason is, no interactions were found between gender, intelligence theories, and overconfidence in [30]’s ([30]) study with undergraduate students. Broadly, those authors found a lack of scholarly knowledge regarding the demographic aspects of overconfident individuals, rather concluding that views of intelligence (entity/fixed or incremental/malleable) and the locus of attention (difficult or easy tasks) may explain self-assessments of performance and effects on learning. On a note of caution, [11] ([11]) found differences in immediate reactions to subjective interpretations of feedback according to gender and race, but their study is not comparable to ours in many ways (e.g., their experimental tasks involved leadership roles, and their focal measurement was the level of importance the participants assigned to those roles); and [65]’s ([65]) study found gender differences in learning performance of students under tutoring schemes, which is also not the case in our study.

We adjusted the experimental computer-aided tasks to the course’s syllabus and the planned activities for the semester. Since the course was about using the computer to support routines and decision-making in organizations, we designed the experimental tasks to include the computer as the tool to make business decisions. We informed the students that they had to be on time to class and stay there until they completed the tasks, as it was a part of their academic evaluations. The tasks involved using an electronic spreadsheet to model and solve decision problems based on the prisoner’s dilemma ([51]; [49]). The problems involved manufacturing companies deciding whether to update their technological infrastructure and how much money to invest (Appendix B). Since streamlining the technological infrastructure is a major competition factor towards production efficiency as well as market share expansion and branding, a company’s decision would have to consider similar strategic choices available to competitors―thus the prisoner dilemma’s rationale.

The experimental model (Figure 1) has several causal paths―some paths for the lab experiment, and other paths for the thought experiment. The reason for such a mixed experimental design is that some causalities are conjectures guiding our reasoning as well as for future empirical research, while other causalities can be actually investigated in the present study, i.e., the effects of feedback on perceptions and actual performance in computer-aided tasks. On the one hand, experiments are usually too limited to help understand human attitudes and behaviors, since any attitude or behavior is too complex to be explained by manipulating a few independent variables and observing their effects through a limited chain of variables. On the other hand, it would be too complicated, if not impossible, to design and perform an experiment with reasonably many variables. Moreover, if the experiment also involves multiple dependent variables, it may become virtually impossible to either do it or develop satisfactorily trustful results. For these and other reasons, scholars have invested in thought experiments to shed light on possible causal chains of events that are too complex to submit to statistical tests whereas certain phenomena require the exploration of multiple causal links even if not currently possible in a fully empirical experiment.

Due to the presence of some conjectures in the experimental design and the number of participants (54 in total), the hypotheses stated before were not submitted to statistical tests. Rather, they were handled with analytical reasoning as the data analysis approach (explained later). Such a decision was aimed at not infringing statistical assumptions that are otherwise required for construct validation. In contrast, our experiment is of an exploratory, interpretive nature and highly dependent on the phenomenological experience of the researchers with the experimental context and the participants. Indeed, the perfect balance between internal validity (consistency) and external validity (generalization) is always a challenge. Our study prioritizes internal validity in many senses (e.g., it includes an arguably true field experiment with high ecological validity and immersion of the researchers in the context and with the participants) while lacking evidence for statistical validity. Other studies do the exact opposite when using an experimental design. For instance, in a study by [14] ([14]) on feedforward interventions, “internal validity was decreased in favor of external validity” and the “random assignment of participants to a control group […] was not possible”.

The experiment involved two sequential tasks. In each task, we first measured the participants’ levels of general computer self-efficacy (G-CSE), tool-specific computer self-efficacy (T-CSE), and problem-specific computer self-efficacy (P-CSE). This three-dimensional CSE construct is an original contribution of our study (the scale we used is available in Appendix A). Afterwards, we asked the students to perform the corresponding task individually and silently. The instructor would not provide any help. After completing each task, the students answered a one-question self-report of performance (SRP_1_, SRP_2_) and the instructor measured their actual task performance (ATP_1_, ATP_2_) with 28 objective criteria (Appendix B). The SRP and ATP measures used a 0–10 scale, which is a popular scale in academic evaluations. Therefore, we collected each student’s performance from the perspective of two stakeholders (SRP by the very student, and ATP by the instructor). This is consistent with the rationale for technology use effectiveness, i.e., effectiveness is dependent on the perspective of each stakeholder ([71]). The ATP measures involved operating the spreadsheet correctly and the quality of the decision tasks. The quality of the decision tasks was easy to measure given that any strategic decision based on the prisoner’s dilemma has a statistically dominant answer―a competitive one ([51]; [49])―due to issues like the agents’ information asymmetry, opportunism, skepticism on the others’ strategic choices, bounded rationality, avoidance of the worst outcome, and acceptance of certain losses. After the first task, we provided feedback on performance to each participant according to a feedback plan (discussed later). Feedback was given in written form, and participants could not exchange information with others about the feedback they received. Students then proceeded to the second task.

As for the environmental contingencies that might impact experiments, we took the risk of not controlling them. Instead, we decided to have a natural task environment, i.e., a regular classroom meeting. Contingencies included the ergonomic factors of the computer use environment (such as temperature, noise, electric power availability, and the quality of office furniture), personal impediments of students to attend the class or be there on time, and other naturally occurring situations. We also did not manipulate the emotional state of participants, such as by changing the room’s visual appeal or engaging in unusual conversation. The intent was to have an ordinary classroom meeting.

The first action in the experiment was the application of a priming procedure for both focusing the students on the task and instructing them on how to model and solve decision problems based on the prisoner’s dilemma (solving such problems was part of the course’s syllabus). Priming was effected with a 10-min video about the dilemma’s principles. The use of video models is an instructional strategy to stimulate learning and self-efficacy ([43]). The experimental actions in our study are summarized in Table 2.

The experiment had the following design ([17]), where R means that the inclusion of individuals in groups followed a random assignment procedure; G_POS_, G_NEG_, and G_CTRL_ are the three groups of feedback; O_1_, O_3_, and O_5_ are the pre-tests for CSE, SRP, and ATP; X_1_ and X_2_ are the experimental treatments (feedback types as described later); “–” is the lack of treatment (neutral feedback); and O_2_, O_4_, and O_6_ are the post-tests for CSE, SRP, and ATP:
RG_POS_O_1_X_1_O_2_RG_NEG_O_3_X_2_O_4_RG_CTRL_O_5_–O_6_

We developed three five-item scales following patterns found in data collection instruments since the first CSE studies, such as those in [21] ([21]), [64] ([64]), and [35] ([35]). Those instruments typically consist of short statements about a specific situation of computer use, sometimes referring to a specific tool as well. We replicated this pattern. However, there are two fundamental differences between our scale and numerous others. First, while some instruments (such as those three mentioned above) employ scales of confidence to measure self-efficacy, we did not find any rationale for using confidence instead of self-efficacy ratings—which are different constructs ([63]). [8] ([8]) in fact discuss how capable, not how confident, one is. Therefore, we asked participants to rate how capable they believed they were to accomplish certain actions on the computer. And a second difference between our scale and extant ones is that ours ranges from 0 to 10 (therefore, it has a neutral point), not from 1 to 10 (with no neutral point). We did not find any argument in favor of not having the neutral point. On the contrary, [21] ([21]) defend it when they mention exactly point number “5” in their scale as the single point representing “moderately confident” individuals, thus suggesting that they see number “5” as a neutral point; however, point number “5” does not stand exactly in the middle of their 1–10 scale. Our instrument is available in Appendix A. It includes three sections, one for each of the CSE types we developed based on the rationale presented in the literature review. The scale was conceptually discussed with our research group in numerous rounds as well as in a version presented at the *Americas Conference on Information Systems* ([72]).

As per the computer-aided tasks, we used two similar designs. We chose similar designs so that we could compare the participants’ CSE, SRP, and ATP across tasks, as well as the effects of feedback. Additionally, having two slightly different tasks would reduce biases in Task #2 (the bias of mere task repetition or, conversely, the bias of having unmatching tasks). To measure ATP, the students received fractional points for each performance criterion fully met, up to 10 possible points per task (Appendix B). A detailed list of real-time verification procedures was also available for the task supervisors (one of the authors and two classroom assistants) to help the execution of the experiment. All students had experience with the scoring procedure, thus we assumed that they were capable of estimating their own performance.

In addition to solving a decision problem, the tasks involved basic computer skills like filling the spreadsheet cells with data, and more complex skills like generating analytical diagrams and formulae. Feedback was given individually in written form to each student after completing Task #1. Three feedback groups were formed (much like the affect manipulation of feedback given to students in [16]): individuals who received positively stated feedback (“Thank you. Your performance was satisfactory.”), individuals who received negatively stated feedback (“Thank you. Your performance was not satisfactory.”), and individuals who received neutral feedback (“Thank you. Let’s begin the second task.”). In selecting the individuals to receive each type of feedback, the instructors analyzed each student’s ATP for Task #1 and then equally distributed each type of feedback (positive, negative, and neutral) to those with actually satisfactory and not satisfactory performance, as follows (Figure 2): half of each type of feedback (positive, negative, and neutral) was randomly given to each of the two performance groups (students with satisfactory and not satisfactory ATP). The criterion for deciding whether a performance was satisfactory or not was the official score to pass an exam in the university (at least 7 points out of a maximum of 10 possible points), but the instructors did not explain to the students the positive or the negative feedback. As such, the participants were free to reflect on all possible performance outcomes (about the general use of the computer, the use of the specific computer tool, or the decision-making activity). As shown later, no student achieved the minimum or the maximum scores possible, thus all feedback messages could be interpreted as meaningful.

In addition to the three valences of affect (positive, negative, and neutral), we modelled feedback as individualized ([75]), absolute ([99]), immediate ([5]), episodic ([53]), unambiguous ([13]; [79]; [11]), and provided by a source (the instructor) that was presumably both credible ([94]) and motivated ([79]). As for the individuals receiving feedback, we had no information about their self-concepts ([58]), orientation to reflection ([4]), or framing of intelligence/abilities ([29]).

On a last note on this study’s originality and rigor, we conducted a thorough search in the recent literature to find similar studies in terms of design and intents. We searched for studies published in the following journals in the field of human–computer interaction: *Computers & Education*, *Computers in Human Behavior*, *Interacting with Computers*, *Behaviour & Information Technology*, and *Information Technology & People*, as well as in all journals of the AIS Senior Scholars’ Basket of Journals[note 3]. The closest studies we found were two experimental studies on the effects of feedback on task performance, and one psychometric study on CSE and self-assessments of performance. First, the study by [10] ([10]) used feedback intervention theory to study the effects of negative, neutral, and positive feedback on task performance as well as interface preferences across individuals, i.e., in a situation of normative feedback. Our study used the same three types of feedback, but we were not interested in comparing individuals due to the person-based framing of technology use effectiveness ([71]). Second, the study by [84] ([84]) measured the impacts of feedback on the subjective cognitive state of participants. Their experimental design was much like ours, in that they had sequential tasks and provided the participants with manipulated positive, negative, and neutral feedback between tasks. However, unlike our study, they intended to compare human- versus computer-generated feedback to verify if there was any difference in the participants’ cognitive reactions. And the third study is one by [1] ([1]) focusing on bias (the direction of judgement error) and accuracy (the magnitude of judgement error) of one’s estimations of CSE and self-assessments of performance. To some extent, our study adds to theirs by including an experimental design to estimate CSE bias and accuracy and by extending their study to adult individuals.

## 4. Results and Discussion

Here, we present the full dataset, the processing of the data, and the findings for the three types of CSE, task performance, and the effects of feedback. The sample was consistently the same in all analyses. Feedback on task performance was the independent variable homogeneously administered to individuals according to the feedback groups (G_POS_, G_NEG_, or G_CTRL_) and irrespectively of actual individual performance. The intent was to verify a variety of individual reactions to feedback. Importantly, no participant knew beforehand that we designed distinct feedback groups. CSE was one of the dependent variables, measured in three domains (G-CSE, T-CSE, and P-CSE). The second dependent variable was task performance, measured both as self-evaluations by the students and as objective evaluations by the instructor. With such a procedure, we addressed a key aspect of technology use effectiveness, i.e., measuring effectiveness from multiple perspectives as it is arbitrarily defined by each stakeholder ([71]).

The data analysis procedures are characterized as analytical reasoning, which is a helpful approach in complex situations or in the lack of statistically significant data. It is not a particular method but rather refers to an ample set of analytical skills that include the insightful, ad hoc elaboration of procedures to solve a pragmatic problem. One example in the IT literature is available in [74]’s ([74]) ex post facto study on the cognitive and behavioral archetype of software developers that correlate with the success of enterprise systems projects. While our study’s sample (54 subjects) is numerically comparable to others (e.g., 46 subjects in [11]), it is not large enough to make us comfortable in performing statistically based comparisons of means. Also, since we coherently did not collect objective data for the thought experiment parts of the experimental design, submitting the available data to statistical tests would not help address those parts. In contrast, by resorting to analytical reasoning, we invested our best analytical efforts to understand and explain the patterns and idiosyncrasies in the data, which are highly dependent on the phenomenological experience of the researchers with the application context and the technology users. Analytical reasoning is in fact expected to produce deep understanding of a situation that would otherwise be constrained within the limits of cold statistical manipulation. For this reason, it is used in U.S. universities[note 4] to develop analytical skills among students and employees. Moreover, the present study was conducted in a highly idiosyncratic society—the Brazilian northeastern region—thus demanding an immersive experience for the researchers to discuss certain attitudinal and behavioral patterns that are possibly present in the data. On a final note, the generalization and replication of the data was not the intention of this study. Rather, it envisioned the elaboration of an experimental design that can be replicated, as well as a phenomenological understanding of attitudes and behaviors in a specific sample of students.

Figure 3 shows what happened in each group for the three measures of CSE. In G_CTRL_, an expressive number of individuals reduced or increased T-CSE and P-CSE, and reduced G-CSE. As G_CTRL_ is the control group, we did not search for an explanation for their attitudinal patterns, but it may result from the individuals understanding the characteristics of the computer-aided tasks after completing Task #1. In G_NEG_, the prevalent pattern was clearly one of reduction in CSE levels, which is coherent with the type of feedback they received. In G_POS_, while many individuals increased their P-CSE in important ways, an expressive number of individuals also decreased G-CSE and T-CSE. G_POS_ individuals may have interpreted the positive feedback as feedback on solving the decision problem with the computer, and not about computer use itself. As such, they showed a pattern for G-CSE and T-CSE that is similar to the pattern in G_CTRL_. Moreover, an increase in P-CSE after positive feedback is coherent with the very concept of technology, which is a means to performing a task and not an end in itself. Therefore, at this moment, we can conclude that individuals, taken as a group, coherently associate feedback on performance with problem solving rather than with technology operation, and they adjust their CSE appraisals accordingly.

Figure 4 shows what happened in each group for average computer self-efficacy (AvCSE), self-reported performance (SRP), and actual task performance (ATP). AvCSE is computed from the three measures of self-efficacy (G-CSE, T-CSE, and P-CSE). The reason for computing AvCSE is to have an estimate on the broad CSE archetype of individuals to compare with their performance (the self-reported and the objectively measured performance). In G_CTRL_, an expressive number of individuals increased SRP and ATP, and other individuals reduced their AvCSE. Again, we did not search for an explanation for the attitudinal patterns in G_CTRL_, but it may be that individuals improved their actual performance (and their perceptions about it) since Task #2 was similar to Task #1. In G_POS_, a vast number of individuals increased their AvCSE, SRP, and ATP. This is arguably due to the reinforcing effects of the positive feedback received, but also to learning across tasks. In G_NEG_, while AvCSE decreased expressively, some individuals did not change their SRP, and others increased their ATP. The increase in ATP was consistent across the feedback groups, possibly due to learning, while SRP mostly remained the same in G_NEG_ possibly due to some individuals being skeptical about the negative feedback they received. As explained before, individuals in a feedback group received the same feedback (whereas they were not aware of the existence of groups), thus the received feedback could be in contrast with SRP or ATP, or both. So far, we conclude that positive feedback correlates with an increase in CSE beliefs and performance, and negative feedback correlates with a decrease in CSE beliefs and an increase in performance.

Now, we focus attention on the individuals (not the groups), particularly the overconfident ones. We are not aware of any method to identify overconfidence; therefore, we relied on experience and on an empirically based, post hoc procedure, as follows: we identified individuals with higher AvCSE than their ATP and selected the individuals with differences between AvCSE and ATP equal or superior to 2.5 points in the scale. Such a magnitude takes into consideration several issues, like the arithmetic difference itself (a very prudent, arguably expressive 25% difference between expectation and effectiveness) and the accuracy, or lack of, with which one is able to assess and report three different types of CSE from which AvCSE is computed. Moreover, it is not any level of CSE that exceeds ATP that should be considered an indication of overconfidence, since moderate–high levels of CSE have been reported as desirable for college students to achieve technology use effectiveness ([73]). Two additional reasons support the proposed heuristic: first, by examining Table 3, one realizes that 2.5 is well above the other AvCSE-ATP differences, except for only six measures between 2.1 and 2.3 reported in the two tasks (out of 108 differences computed in total), meaning that 2.5 is significantly above 94% of all AvCSE-ATP differences; and the second reason is that a range of 2.5 is also well above the difference needed for a student to move between grades in the Grade Point Average (GPA) system[note 5]. Therefore, 2.5 and above can be reasonably accepted as a manifestation of overconfidence. On a final note, such a rationale does not mean that difference scores below 2.5 may not represent overconfidence too, but that 2.5 and above is a safe threshold for our analyses.

Table 3 shows the overconfidence cases we considered. Students are identified according to their classroom section (“M” for the morning, “E” for the evening), and overconfidence levels are colored in black. We identified 17 overconfident students in the first task, and five students in the second task, i.e., one-third of the participants were initially overconfident based on our conservative heuristic for overconfidence. Such a proportion is in line with a study with junior-high-school students about their digital competencies, which has found that “only a few of participants’ perceived skills were related to their actual performance […] participants displayed high confidence in their digital literacies and significantly over-estimated their actual competencies” ([70]). Another study by [1] ([1]) found similar results among primary school students regarding an overestimation of competencies to use the information and communication technologies.

In G_CTRL_, four individuals showed overconfidence in Task #1, and all of them reduced or eliminated overconfidence in Task #2. This group also had the only individual (M25) with initially acceptable high confidence (1.9) and overconfidence (3.03) after receiving (neutral) feedback. That is, in no other situation did individuals with high confidence in Task #1 show overconfidence in Task #2. The difference between this single case and the other four in G_CTRL_ is that the four overconfident individuals in Task #1 had low or borderline ATP, so they seem to have naturally adjusted their CSE for Task #2 while also improving their performance due to learning from Task #1.

In G_POS_, positive feedback generally reinforced the individuals’ CSE beliefs, to the extent that two of the four overconfident individuals in Task #1 remained overconfident in Task #2. One of the four individuals (M12) adjusted his or her CSE to a more acceptable level, while the individual with the highest overconfidence of all (E20) probably realized how far he or she was to performing well regardless of the positive feedback received and reacted accordingly.

In G_NEG_, we found the largest number of individuals (nine) with overconfidence. Since the individuals were randomly assigned to groups according to a balanced procedure of matching types of feedback to actual performance (Table 3), and since we controlled for the presence of a balanced number of morning- and evening-section students in each group as well as for pairing the most relevant demographic variables, we do not know why this happened, other than knowing that the number of individuals in each group (18) was relatively small to assure a fully homogeneous sampling across the groups. One of the nine overconfident G_NEG_ students (M9) refused to complete Task #2 after receiving the negative feedback on Task #1. That student’s behavior cannot be explained as disappointment with the negative feedback received, given that the student achieved only 5.25 evaluation points out of 10 in Task #1, i.e., the student’s actual performance was below the course’s passing level (7.0). The student left the classroom and did not explain why. In the remaining eight G_NEG_ overconfident cases, six students adjusted their CSE levels downwards and more coherently with ATP levels, but two students were not able to sufficiently improve their ATP in Task #2 so as to avoid being considered overconfident again. Those who adjusted their CSE to more realistic levels may hold incremental views of intelligence (those who believe that people can improve their abilities), thus being open to learning with negative absolute feedback ([99]).

We were intrigued by the unexpected fact that one student withdrew from the experiment; therefore, we tried to understand it using theories of feedback. In addition to a general understanding that feedback activates emotional responses ([50]), we know that entity-intelligence people (those who believe that ability is fixed/stable) are likely to reject negative feedback ([22]) and avoid tasks that might require reflection on low performance ([30]). Therefore, this may explain that student’s behavior as well as why other students had difficulty in adjusting their CSE levels.

Overall, we concluded that (1) CSE adjustments and increases in ATP occurred partially as a natural learning process across tasks as the three groups of feedback showed reasonably similar patterns when adjusting their CSE and ATP levels; (2) positive feedback contributed to reinforcing CSE beliefs and increasing ATP; and (3) negative feedback contributed to adjusting overestimations of CSE more than increasing ATP. However, as per the first conclusion, feedback on task outcomes (as modelled in our study) usually limits the learning process ([50]); therefore, more research is needed to isolate the effects of either learning or feedback. Also, at the individual level, neutral feedback (no real feedback) seems to have been as powerful as positive/negative feedback when adjusting CSE and ATP levels for the particular case of overconfident individuals while not causing emerging overconfidence in other individuals. We did not analyze, however, the possible emergence of underconfidence. Such results are in line with [16] ([16]), who found that neutral feedback given to secondary school students, as compared with emotionally reinforcing feedback, was more effective. Also, in interviews with early-career academic doctors, [94] ([94]) were inconclusive on whether the valence of feedback is influential over one’s reactions in different motivation scenarios.

## 5. Implications for Theory and Practice

This study has implications for scholarly knowledge. First, we have offered an original, exploratory decomposition of CSE in three dimensions (general, problem-specific, and tool-specific). Second, we offered a conceptualization and heuristic to identify overconfidence―as a post hoc measure of unrealistically high self-efficacy beliefs. Third, this study is among the few empirical ones on the framing of technology use effectiveness, i.e., the conception of effectiveness as a stakeholder’s arbitrary perspective. Fourth, this is one of few studies, if any, discussing the need to integrate attitudes (one’s CSE and SRP levels as well as the motivational reactions to feedback), theoretical abilities (one’s previous experience with a course’s contents as well as learning on the task), and practical skills (one’s actual behaviors towards problem modeling and solving) to explain technology use effectiveness. Fifth, this is among the very few studies in the human–computer interaction domain to articulate thought and lab experiments and to analyze experimental data with a qualitative approach.

This study also has implications for classroom activities, organizational hiring, training, and team building. In summary, we provide three new scales for CSE (Appendix A), two versions of a scenario-based computer-aided decision task for use in the classroom and in professional training (Appendix B), an experimental design, and numerous insights into the significant numbers of overconfident individuals that may be present in a work group (indeed, we found around one-third of individuals to be overconfident in our sample, by using a very conservative measure). Particularly helpful is the use of the tool in Appendix B. When teaching topics like the commoditization of IT ([18]), the productivity paradox ([2]), and business competition in the IT industry, students want to see real situations to better understand the logic of the prisoner’s dilemma in action. The tool we offer can be used in this sense for a first glimpse at the concepts and how decisions are made when a company’s strategy needs to take into account a competitor’s strategy. Another possible use of the tool, along with the CSE instrument in Appendix A, is to promote self-reflection by the very participants of behavioral experiments and identify individual decision-making patterns ([32]). Such convenient experiments may efficiently reveal the presence of overconfident individuals in a team, as expectations placed on those individuals, and by them, may not correspond to reality and thus result in frustration, opposition, and the discontinuance of work (such as what happened with one overconfident student who abandoned our experiment).

## 6. Limitations and Future Research

This study has limitations. The first limitation is the quantity of individuals in each experimental group (three groups of 18 individuals each, 54 in total). We had the opportunity to study the individuals in an almost natural setting of decision-making under performance assessment (students under real assessment by an instructor), but the drawback was that we had fewer individuals than needed to study certain statistics. While we have found statistically based experimental studies with fewer individuals than ours (e.g., 46 individuals in [11]), we saw more problems than benefits in doing so. Therefore, we opted for an analytical reasoning approach, which provided high ecological validity due to the various aspects of how the experiment was designed and conducted as well as the opportunity to reevaluate established concepts ([52]). We want to take the opportunity to comment a bit further on the issue of qualitative versus statistically based analyses. No sufficient explanation exists for two events being causally connected—neither philosophy nor statistics can solve this fact. Multivariate statistics is based on correlations, and there is always the possibility that two events are spuriously correlated either because they are the concomitant consequences of a common cause or due to pure chance. Therefore, all we can do before better knowledge is available in a given situation (either with or without statistically sound measures) is to develop a temporarily satisfactory explanation based on our own factual experience and judgement about how and why the events of our interest manifest as they do. In the present study, we have opted to invest our best efforts in analytical reasoning rather than having our conclusions constrained within the statistically accepted boundaries. If we had a larger sample size and had developed full statistical measures, our discussion would inevitably focus on them, thus causing us to lose the opportunity to conduct the in-depth, phenomenological analyses we actually performed.

Another limitation of this study is that, if we wanted to understand our data according to individual user profiles, we do not have information about certain personal characteristics of each individual, such as their self-concepts, orientation to reflection, and the framing of their intelligence/abilities. But again, this stems from the opportunity we had to study individuals in a natural setting. Even if we could benefit from understanding how overconfident individuals react to affect-based feedback according to individual characteristics, we would rather search for an answer to the broader question on the effects of feedback over one’s possibly inflated self-perceptions of performance before and after performing similar tasks on the computer. Also, we strongly argue in favor of a more realistic, ecologically valid experimental setting such as ours rather than processing large statistical data from artificially designed environments or self-reported attitudinal surveys. Still concerning an individuals’ traits, we also did not consider if some participants had a significant inclination towards response biases such as social desirability ([31]) and acquiescence ([95]). While some may see this as a limitation[note 6], we had not considered such biases as intervening factors because we had not asked respondents to express their views about socially shared issues.

A third limitation is the heuristics we used to analyze the data, since heuristics are always questionable for their essentially empirical rationale. A fourth limitation[note 7] is that we had not considered the possibility that the students in the morning section could inform those in the evening section about the experiment. In the university where the study took place, most students take classes independently and work for the remaining hours of the day, thus it is unlikely that they would have strong reasons to immediately share information with others about the activities in the classroom. Also, we did not inform the first classroom that the same exercise would be performed again with other students in the same day. Fifth, we did not manipulate any of the specific CSE types (G-CSE, T-CSE, P-CSE). While this may be a limitation, it is coherent with the fact that we focused on average CSE (AvCSE) when comparing the CSE levels with actual task performance (ATP). A sixth limitation—although not directly related to the research question—is that we did not analyze whether the experimental treatments caused the emergence of underconfidence in some individuals.

Besides addressing the limitations mentioned before, we expect that future research will refine the instruments to measure our newly developed three-dimensional CSE construct. Future research may also contribute to the long-lived debate on whether self-efficacy beliefs and affective feedback correlate with individual performance in computer-aided tasks. Another avenue for future research is to compare the outcomes of learning from feedback with the outcomes of learning from pure experience. Still regarding the sources of learning and self-efficacy gains, future research can test the effects of the use of priming to introduce a decision problem. In our study, priming was used to level the participants’ awareness of the tasks to be completed, but it is possible that the model–observer similarity hypothesis and the task appropriateness hypothesis explain the cognitive gains and losses of some individuals due to personal identification with the priming stimulus or with those who produced it ([81]; [42], [43]). Another suggestion for future studies is to deepen our understanding about the thought-experiment causal paths, hopefully developing instruments to measure them. Last, the present study can be helpful for research on the reactions to feedback within the self-motives literature. The present study was developed with participants who were in a passive position with regard to the feedback process, i.e., the design and the purpose of the feedback were based on the interests of another stakeholder (the instructors/researchers). However, there is additional literature on people who actively seek for, and react to, feedback due to self-enhancement and self-improvement interests ([3])[note 8].

## 7. Conclusions

This study aimed to understand the reaction of overconfident students to feedback received from their instructors on task performance under the premises of technology use effectiveness, i.e., the effectiveness with which an individual deploys the needed resources to achieve an arbitrarily defined technology use purpose. To answer the research question, we developed a novel, three-dimensional measure of CSE, a scenario-based tool to conduct experiments with sequential decision tasks in the classroom, and a mixed, interpretive experiment on the effects of feedback over one’s CSE beliefs and performance in computer-aided decision-making. We found that the valence of feedback may not be decisive―when compared with neutral feedback―in how individuals elaborate their CSE beliefs and perform decision tasks on the computer if skills and learning are expected to mature naturally. Therefore, our study adds to a body of research that does not find clear benefits stemming from feedback in general and for overconfident individuals in particular. This is intriguing as common sense suggests feedback is important to have “one’s feet on the ground” and achieve task effectiveness.

## Figures and Tables

**Figure 1 behavsci-15-00511-f001:**
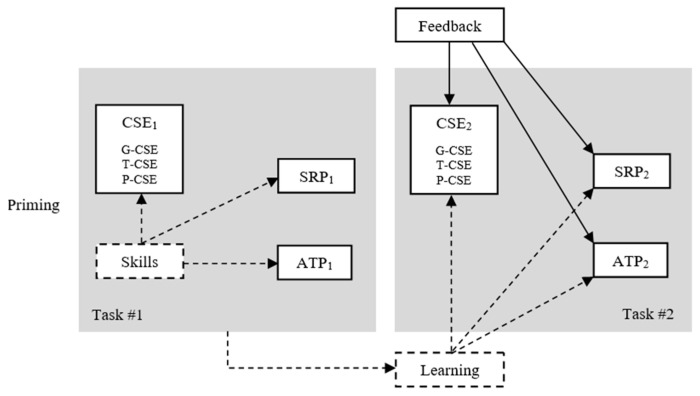
The experiment. Notes: dashed lines represent the thought experiment’s causal paths and latent variables; full lines represent the lab experiment’s causal paths and variables that were effectively measured. The model integrates motivational attitudes (the levels of CSE and SRP, and the reactions to feedback), theoretical abilities (previous experience with the course’s contents along with the modelling instructions received from the priming procedure), and practical skills (the actual skills impacting the formation of CSE and ATP).

**Figure 2 behavsci-15-00511-f002:**
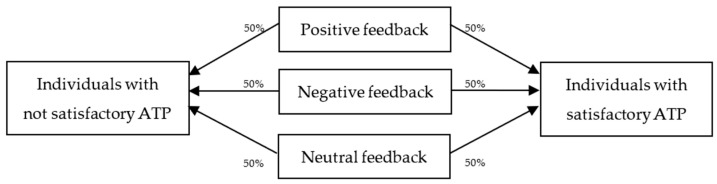
Feedback plan.

**Figure 3 behavsci-15-00511-f003:**
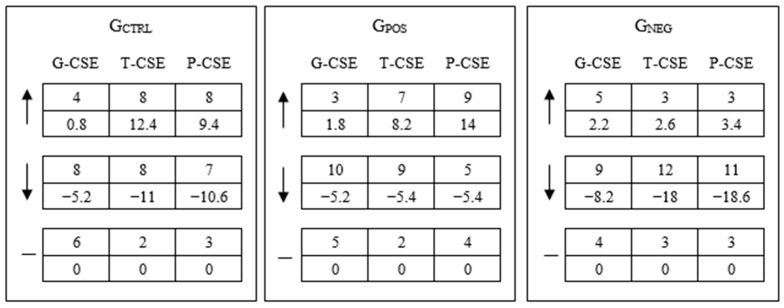
Computer self-efficacy at the general, tool, and problem levels. Note: Numbers in the first rows represent the quantity of individuals with increased (“↑”), decreased (“↓”), or equal (“—”) measures after feedback, and numbers in the second rows represent the accumulated differences regarding the first (Task #1) and second (Task #2) measures of CSE from all individuals in the corresponding column. One individual in G_NEG_ did not complete Task #2 and did not provide data on P-CSE.

**Figure 4 behavsci-15-00511-f004:**
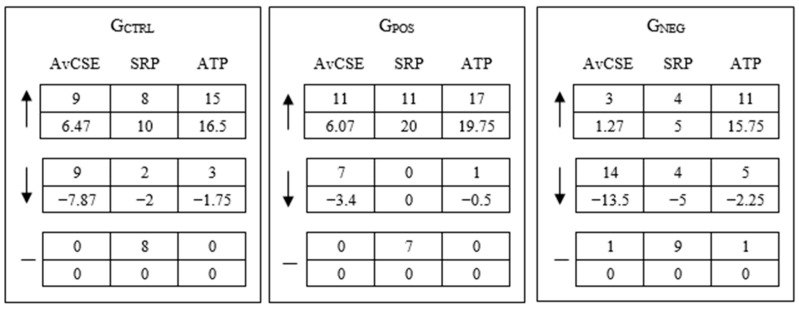
Average self-efficacy, self-reported performance, and actual task performance. Note: Numbers in the first rows represent the quantity of individuals with increased (“↑”), decreased (“↓”), or equal (“—”) measures after feedback, and numbers in the second rows represent the accumulated differences regarding the first (Task #1) and second (Task #2) measures of CSE from all individuals in the corresponding column. One individual in G_NEG_ did not complete Task #2 and, as such, did not provide data on SRP and ATP.

**Table 1 behavsci-15-00511-t001:** The experimental groups.

Demographic Variable	G_CTRL_	G_POS_	G_NEG_
Participants (quantity)	18	18	18
Second semester participants (%)	89	94	89
Average age (years)	21.8	23.7	22.7
Average first contact with computers (year)	2002	2000	2002
Participants with experience in computer-related industry internship (quantity)	4	2	3
Participants with computer-related work experience (quantity)	11	8	13
Average duration of internship (months)	7	17.5	7
Average work experience (months)	31	86	51
Female participants (%)	55	72	28
Male participants (%)	45	28	72
Average importance attributed to computer use for personal issues (0–10)	8.83	8.78	8.67
Average importance attributed to computer use for professional issues (0–10)	9.5	8.94	9.44

**Table 2 behavsci-15-00511-t002:** Actions in the experiment.

Sequential Action	Code	Duration (min)
Priming with a video about the prisoner’s dilemma	Priming	10
Measurement of computer self-efficacy (G-CSE, P-CSE, T-CSE)	CSE_1_	5
Electronic spreadsheet task (first version)	Task #1	35
Self-evaluation of performance	SRP_1_	*
External evaluation of actual performance	ATP_1_	**
Feedback intervention	Feedback	***
Measurement of computer self-efficacy (G-CSE, P-CSE, T-CSE)	CSE_2_	5
Electronic spreadsheet task (second version)	Task #2	35
Self-evaluation of performance	SRP_2_	*
External evaluation of actual performance	ATP_2_	**

Notes: * Self-evaluation was part of the task, i.e., it was completed within the 35 min task limit. ** Actual task performance was evaluated as soon as each student finished the corresponding task. *** Feedback was given immediately to each student after Task #1.

**Table 3 behavsci-15-00511-t003:** Overconfident individuals.

**Student (G_CTRL_)**	**AvCSE_1_**	**ATP_1_**	**AvCSE_1_—ATP_1_**	**AvCSE_2_**	**ATP_2_**	**AvCSE_2_—ATP_2_**
M2	9.07	5.50	3.57	9.00	7.00	2.00
M4	4.73	5.75	−1.02	6.20	6.25	−0.05
M10	4.07	2.00	2.07	4.67	4.00	0.67
M13	2.67	3.00	−0.33	3.27	3.50	−0.23
M18	7.00	3.50	3.50	3.20	4.75	−1.55
M25	9.40	7.50	1.90	9.53	6.50	3.03
M27	5.60	5.00	0.60	5.67	5.50	0.17
M28	9.33	6.75	2.58	8.87	8.75	0.12
M29	5.73	3.00	2.73	5.60	3.75	1.85
E1	6.80	5.50	1.30	6.47	7.00	−0.53
E3	7.87	7.00	0.87	8.40	8.00	0.40
E6	5.93	5.75	0.18	8.13	6.00	2.13
E7	5.40	5.75	−0.35	3.53	5.50	−1.97
E15	8.27	6.50	1.77	7.93	6.00	1.93
E16	8.60	7.75	0.85	9.00	9.50	−0.50
E18	4.60	6.75	−2.15	4.40	8.75	−4.35
E23	8.80	8.75	0.05	9.27	9.00	0.27
E25	3.53	4.75	−1.22	2.87	5.50	−2.63
**Student (G_POS_)**	**AvCSE_1_**	**ATP_1_**	**AvCSE_1_—ATP_1_**	**AvCSE_2_**	**ATP_2_**	**AvCSE_2_—ATP_2_**
M6	8.27	5.75	2.52	8.87	6.25	2.62
M7	3.13	5.00	−1.87	2.07	5.75	−3.68
M8	3.53	6.50	−2.97	4.47	8.00	−3.53
M12	6.80	4.00	2.80	6.60	5.25	1.35
M15	2.80	4.75	−1.95	3.00	6.50	−3.50
M17	9.07	9.00	0.07	8.60	9.75	−1.15
M20	8.73	5.75	2.98	8.87	6.00	2.87
M22	7.00	6.00	1.00	7.07	7.00	0.07
M23	0.73	4.25	−3.52	1.73	5.25	−3.52
M24	7.67	7.25	0.42	7.87	7.50	0.37
E2	4.67	6.00	−1.33	4.47	6.75	−2.28
E4	6.33	6.00	0.33	6.07	8.00	−1.93
E10	5.07	3.75	1.32	6.00	4.00	2.00
E11	5.33	6.50	−1.17	6.07	7.75	−1.68
E12	3.80	4.00	−0.20	3.07	3.50	−0.43
E14	6.07	7.50	−1.43	6.73	7.75	−1.02
E20	9.33	4.00	5.33	8.87	9.25	−0.38
E24	4.67	5.25	−0.58	5.27	6.25	−0.98
**Student (G_NEG_)**	**AvCSE_1_**	**ATP_1_**	**AvCSE_1_—ATP_1_**	**AvCSE_2_**	**ATP_2_**	**AvCSE_2_—ATP_2_**
M9	9.33	5.25	4.08	10.00	−	−
M1	7.27	7.75	−0.48	7.20	8.25	−1.05
M3	8.73	6.50	2.23	8.53	7.75	0.78
M5	7.87	5.00	2.87	7.47	4.50	2.97
M11	8.47	5.00	3.47	7.20	7.50	−0.30
M14	6.07	6.00	0.07	5.60	5.50	0.10
M16	5.13	6.25	−1.12	4.60	6.00	−1.40
M19	8.60	7.25	1.35	8.20	9.50	−1.30
M21	6.53	8.25	−1.72	6.80	8.75	−1.95
M26	2.73	4.75	−2.02	2.67	6.25	−3.58
E5	8.60	7.00	1.60	8.60	8.25	0.35
E8	7.20	4.75	2.45	3.93	4.75	−0.82
E9	8.00	5.50	2.50	5.93	5.75	0.18
E13	5.87	1.75	4.12	3.33	1.00	2.33
E17	9.00	6.50	2.50	8.33	7.75	0.58
E19	5.53	3.00	2.53	4.13	6.50	−2.37
E21	8.60	5.25	3.35	8.93	6.25	2.68
E22	9.67	8.75	0.92	9.53	8.50	1.03

Notes: Overconfidence levels are colored in black, and non-overconfidence levels are colored in grey. Student M9 left the room after filling the CSE_2_ form with two maximum scores. Student E8 was a borderline case considering overconfidence.

## Data Availability

Data is contained within the article.

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
