# Peer review of "Computer Self-Efficacy and Reactions to Feedback: Reopening the Debate in an Interpretive Experiment with Overconfident Students"

_behavsci, 2025, doi:10.3390/bs15040511_

Round 1

Reviewer 1 Report

Comments and Suggestions for Authors

I had the opportunity to review your submission and have read through your paper carefully. I see the potential of your work, and the possible added value of your topic to the literature.  Yet, I do have several substantial concerns and critique, I also added some advice and sources.

To begin with, I believe that the structure of the literature review could be more organized and have a clearer objective. Currently, it presents a large amount of enumerative information, but I would prefer to see it structured in a more academically rigorous way. Moreover, the operationalization of ‘overconfidence’ was not clearly explained. There is little concrete scientific evidence to support the assumptions of this study: the literature review cites a considerable number of sources, some of which do not appear to be directly relevant to this study. At the same time, there are other pertinent sources that could add depth to the development of your literature overview. Below are some suggested references that I believe would be relevant:

Regarding 'the self' and how feedback affects it:

  • Anseel, F., Strauss, K., & Lievens, F. (2017). How future work selves guide feedback seeking and feedback responding at work. In Ferris, L., Johnson, R.E., & Sedikides C. (Eds.). The Self at Work (pp. 294-318). New York: Routledge.

Papers on self-motives: People require feedback on their performance to work effectively, but this can sometimes be challenging due to the confrontation between a positive self-image and reality. Two motives play a role: the desire to feel good about oneself (self-enhancement) and the desire to improve oneself (self-improvement), which can sometimes lead to a paradox. This could be further clarified. Here are some suggested sources:

  • Anseel, F., Lievens, F., & Levy, P. E. (2007). A self-motives perspective on feedback-seeking behavior: Linking organizational behavior and social psychology research. International Journal of Management Reviews, 9(3), 211-236.
  • Jordan, A. H., & Audia, P. G. (2012). Self-enhancement and learning from performance feedback. Academy of Management Review, 37(2), 211-231.

Additional sources on feedforward and the importance of looking ahead:

  • Budworth, M. H., Latham, G. P., & Manroop, L. (2015). Looking forward to performance improvement: A field test of the feedforward interview for performance management. Human Resource Management, 54(1), 45-54.
  • Kluger, A. N., & Nir, D. (2010). The feedforward interview. Human Resource Management Review, 20(3), 235-246.

Concerning your methodology, I found it a creative approach to combine thought and lab experiments. However, the rationale for this combination should be further elaborated. The paper currently provides brief reasoning, but it could be more thoroughly explained—why precisely is the combination with thought experiments important?

In the paper, you state: "We informed the participants about the experiment only after its conclusion, when we asked for their individual consent to analyze and publish the data." This implies that participants were unaware they were taking part in an experiment beforehand. Do you consider this ethically sound, did you coordinate this with the ethics committee of your university? Additionally, was there a risk that participants from the first session informed those in later sessions about the nature of the experiment?

You also mention in the paper that “This was done with a 10-minute video about that dilemma’s principles.” How did you verify that participants actually watched the video and understood the instructions? What was your manipulation check? Also the check for feedback: Has it been verified whether each student actually read the feedback? After all, it is not possible to determine which students genuinely reflected on the feedback and which did not.

How did the authors determine the threshold of a 2.5-point (or more) difference between AvCSE and ATP for classifying an individual as overconfident?

Furthermore, you write that “We developed three five-item scales based on Compeau and Higgins (1995), Murphy et al. (1989), and Gist, Schwoerer, and Rosen (1989), with scores ranging from 0 to 10 to measure the three types of CSE we originally proposed.” How were these scales developed? More information is needed on this point, possibly in a supplemental section.

Next, you explain that “The independent variable was homogeneously administered to individuals according to the feedback groups (GPOS, GNEG, or GCTRL) and irrespective of actual individual performance.” This should be made clearer—does this mean that even high-performing students (who felt they had performed well) could receive negative feedback? Could this not be a confounding factor affecting the study’s results? Sidenote: There are clear differences between the groups Gctrl, Gpos, and Gneg (e.g., gender, internship experience, etc.). Could these differences have influenced the results? Can you clarify this more?

You also mention this: “Therefore, this may explain that student’s behavior as well as why other students had difficulty adjusting their CSE levels. Since that student was a woman, another possible explanation, according to a review on self-perceptions by Chevrier et al. (2020), relates to female college students being more self-critical about their academic abilities and more affected by self-esteem issues that address their intellectual abilities, scholastic competence, and expectations about college life.” This explanation seems somewhat reductionist—it is based on a single case and does not seem entirely relevant to the study’s broader scope, particularly since gender is stated elsewhere as not being a relevant variable.

Discussion part:

However, the most significant issue with the study, in my view, is the sample size. The authors acknowledge this in the discussion: “Therefore, all we can do before better knowledge is available in a given situation (either with or without statistically sound measures) is to develop a temporarily satisfactory explanation based on our own factual experience and judgment about how and why the events of our interest manifest as they do. In the present study, we have opted to invest our best efforts in analytical reasoning rather than having our conclusions constrained within the statistically accepted boundaries.” A stronger justification is required—perhaps by referencing more other studies that worked with similarly small samples or by drawing on relevant literature to reinforce the methodological approach. A more robust argument (either methodological or literature-based) is necessary to justify the small sample size.

Differences are also observed in the control group, both in ATP and AvCSE. How can these differences be explained? This is currently not considered in the conclusions.

The practical implications could be more specifically articulated. As they stand, they remain somewhat vague in terms of what the study offers in concrete terms. The utility of the final implication (the tool) is also not entirely clear.

In the final conclusion, the authors state: “This study aimed at understanding the reaction of overconfident students to feedback received from their instructors on task performance under the premises of technology use effectiveness.” This appears to be a rather narrow contribution—what is the broader significance of understanding this? The value of the paper likely extends beyond this and should be more explicitly emphasized.

Author Response

(f needed, please see attached also the response to the other Reviewer.)

Dear Reviewer:

We write this revision letter regarding our manuscript code “behavsci-3473350” (originally titled “Computer Self-efficacy and Reactions to Feedback in the Classroom: Reopening the Debate in an Interpretive Experiment with Overconfident Students”). We are pleased to know that the Editor and the Reviewers found merits in the manuscript and see a possibly good future for it after a revision.

We are always happy to receive reviews as they make it perfectly clear how important the peer review process is and how true it is that multiple and acceptable perspectives exist about any work. The two reviews we received showed to us the need to improve our communication to the reader in many aspects, as well as that our work is limited by additional factors besides those we had initially identified. We thus carefully revisited the paper in full and paid particular attention to addressing the review comments both in the revised paper and in this response letter. Here, we discuss each review comment as well as a few additional opportunities for improvement that we identified and addressed during the revision.

Overall, the Reviewers found issues in need of improvement in terms of the study’s conceptual and methodological aspects. While the timeframe for this revision is a tight one (only 10 days), we believe to have solved most issues by better positioning our study within the epistemological spectrum and communicating it properly to the reader. In the next pages, we describe in due detail the Reviewers’ comments and how we addressed them in the revised paper. We have also uploaded, to the manuscript handling system, a file with all revision marks.

----

REVIEWER #1

I had the opportunity to review your submission and have read through your paper carefully. I see the potential of your work, and the possible added value of your topic to the literature.  Yet, I do have several substantial concerns and critique, I also added some advice and sources.

AUTHORS

Thank you for the encouraging remarks. We do hope to have addressed your concerns with the needed quality as commented below.

----

 REVIEWER #1

To begin with, I believe that the structure of the literature review could be more organized and have a clearer objective. Currently, it presents a large amount of enumerative information, but I would prefer to see it structured in a more academically rigorous way.

AUTHORS

This comment caught us surprised, especially regarding the critique on the academic rigor. We are scholars from the fields of consumer behavior, human-computer interaction, and organization studies, and we believe the academic genre in these fields may be the reason for the Reviewer’s comment, as there may be a difference from that in the field of behavioral sciences. We read articles from Behavioral Sciences (journal) before this submission, and we realized just that. We found the articles in Behavioral Sciences much more concise, some of them even not including a Literature Review section. We are afraid that we will not be able to change dramatically how our literature review is organized, since we simply would not know how to do it as per our scholarly practice, besides the extremely tight deadline of just 10 days. The background for our study comes from the fields of Human Resources Management and Human-Computer Interaction, where articles are organized like ours. In these fields, we work with constructs (phenomena that are not directly measurable and that are mostly based on agreed-upon constructions by scholars), thus it is usually necessary to remind readers about their concepts, research traditions, and current findings. This is the case for our constructs (computer self-efficacy, task performance, feedback interventions, and learning). By the way, one of the contributions of this paper is exactly to provide a reasonably deep review on those constructs.

Anyway, as the Reviewer will see, we made numerous interventions in the text to make it more appealing to the reader. Also, we are open to discuss this issue further.

----

REVIEWER #1

Moreover, the operationalization of ‘overconfidence’ was not clearly explained.

AUTHORS

As we state in the paper, “[w]e are not aware of any method to identify overconfident individuals, thus we relied on experience and on an empirically based, post-hoc procedure”. We thus developed a heuristic to identify overconfident individuals, as discussed in the text. However, if there is another method in the literature to identify overconfident individuals, we will be happy to consider it. In the revised text, we provided additional details about the heuristic, as follows:

“We are not aware of any method to identify overconfident individuals, thus we relied on experience and on an empirically based, post-hoc procedure: we identified individuals with higher AvCSE than their ATP and selected the individuals with differences between AvCSE and ATP equal or superior to 2.5 points in the scale. Such a magnitude takes into consideration several issues, like the arithmetic difference itself (a very prudent, arguably expressive 25% difference between expectation and effectiveness) and the accuracy, or lack of, with which one is able to assess and report three different types of CSE from which AvCSE is computed. Moreover, it is not any level of CSE that exceeds ATP that should be considered an indication of overconfidence, since moderate-high levels of CSE have been reported as desirable for college students to achieve technology use effectiveness (Porto-Bellini et al., 2016). Two additional reasons support our heuristic: first, by examining Table 3, one realizes that 2.5 is well above the other AvCSE-ATP differences, except for only six measures between 2.1 and 2.3 reported in the two tasks (out of 108 differences computed in total), meaning that 2.5 is significantly above 94% of all AvCSE-ATP differences; and the second reason is that a range of 2.5 is also well above the difference needed for a student to move between grades in the Grade Point Average (GPA) system. Therefore, 2.5 and above can be reasonably accepted as a manifestation of overconfidence. On a final note, such a rationale does not mean that difference scores below 2.5 may not represent overconfidence too. However, since this study is not a statistical one, there are no implications for not considering the below-2.5 individuals in the analysis.”

----

REVIEWER #1

There is little concrete scientific evidence to support the assumptions of this study: the literature review cites a considerable number of sources, some of which do not appear to be directly relevant to this study.

AUTHORS

The Reviewer does not specify the assumptions, thus we searched for all occurrences of “assumption” and related terms in our text and found the ones below. We comment how we handled each one.

  • Excerpt: “The main assumption is that we should know better how a user conceives his or her actual competencies to operate a system according to stated purposes in order to realize the presence and the impact of possible false beliefs on what is needed to perform computer tasks.”

Our comment: We excluded this sentence from the Introduction section, since it is tautological. It does not add any new insight.

  • Excerpt: “CSE is still a construct under conceptual refinement and empirical validation. With such an assumption (…)”

Our comment: This sentence comes from the previous historical account we provided on the CSE construct. However, the word “assumption” does not add anything really useful, thus we reorganized the sentence as follows: “It is thus apparent that CSE is still a construct under conceptual refinement and empirical validation. We advocate that, in the light of technology use effectiveness, the concept of CSE should (…)”.

  • Excerpt: “Kluger and DeNisi (1996) developed feedback intervention theory motivated by about 90 years of poor scholarly knowledge on how feedback relates to behavior and performance, as well as by evidence that feedback significantly reduces performance in certain situations or does not have any effect at all—whereas the widely accepted assumption is that feedback improves performance (...) the simplistic assumption that any behavior could be regulated by feedback.”

Our comment: Those assumptions were developed by Kluger and DeNisi, not by us.

  • Excerpt: “… part of our model can be tested in a lab experiment, and the other part is made of assumptions and reasoning.”

Our comment: We administered only one treatment to the participants of the experiment, i.e., the feedback they received on individual performance after the first computer-aided task. Therefore, we modeled all the other parts of the experiment as a thought experiment. Anyway, we excluded this full sentence, since the sentence that precedes it was already fully informative (“Our experiment employs both forms of experimental design due to the presence of testable and latent causal paths.”).

  • Excerpt: “… we studied technology use with the theoretical lenses of use effectiveness, which requires the definition of the purposes of use and assumes that more than one perspective of use effectiveness may exist.”

Our comment: The assumption comes from the mentioned source on technology use effectiveness.

  • Excerpt: “… those authors found a lack of scholarly knowledge regarding the demographic aspects of overconfident individuals, rather assuming that views of intelligence (entity/fixed or incremental/malleable) and the locus of attention (difficult or easy tasks) may explain self-assessments of performance and effects on learning.”

Our comment: The assumption comes from the mentioned source on the framings of intelligence.

  • Excerpt: “All participants had experience with the maximum scores per activity in each task, so that we assumed they had the ability to estimate their performance.”

Our comment: This is not a theoretical assumption, but a practical one to justify that the participants in our task experiment had experience with similar tasks. We believe it can be accepted as is by the reader, but we reworded it as: “All students had experience with the scoring procedure, thus we assumed that they were capable of estimating their own performance.”

  • Excerpt: “… since learning across tasks can be assumed to have occurred also in GPOS and GNEG (…)”

Our comment: This is a reasonable assumption, i.e., that learning occurs across tasks. It is an assumption (and not a testable/tested fact) as we had not manipulated it in the experiment. Anyway, we excluded that sentence as we improved the communication of the full paragraph.

----

REVIEWER #1

At the same time, there are other pertinent sources that could add depth to the development of your literature overview. Below are some suggested references that I believe would be relevant: Regarding 'the self' and how feedback affects it:

Anseel, F., Strauss, K., & Lievens, F. (2017). How future work selves guide feedback seeking and feedback responding at work. In Ferris, L., Johnson, R.E., & Sedikides C. (Eds.). The Self at Work (pp. 294-318). New York: Routledge.

Papers on self-motives: People require feedback on their performance to work effectively, but this can sometimes be challenging due to the confrontation between a positive self-image and reality. Two motives play a role: the desire to feel good about oneself (self-enhancement) and the desire to improve oneself (self-improvement), which can sometimes lead to a paradox. This could be further clarified. Here are some suggested sources:

Anseel, F., Lievens, F., & Levy, P. E. (2007). A self-motives perspective on feedback-seeking behavior: Linking organizational behavior and social psychology research. International Journal of Management Reviews, 9(3), 211-236.

Jordan, A. H., & Audia, P. G. (2012). Self-enhancement and learning from performance feedback. Academy of Management Review, 37(2), 211-231.

Additional sources on feedforward and the importance of looking ahead:

Budworth, M. H., Latham, G. P., & Manroop, L. (2015). Looking forward to performance improvement: A field test of the feedforward interview for performance management. Human Resource Management, 54(1), 45-54.

Kluger, A. N., & Nir, D. (2010). The feedforward interview. Human Resource Management Review, 20(3), 235-246.

AUTHORS

We thank the Reviewer for such remarks. In the first version of the paper, we have addressed this issue superficially. Now we added several notes on the two most important concepts mentioned in the suggested sources: feedforward and feedback seeking. We added the following paragraphs:

LITERATURE REVIEW:

“(…) feedback may be discussed in contrast with an intervention called feedforward and whether an individual passively receives or actively seeks feedback*. Feedforward refers to one’s intentions for the future rather than to performance in the past (Budworth, Latham, & Manroop, 2015), while feedback seeking refers to a change in the locus of control in the feedback process (Anseel, Lievens, & Levy, 2007). In our study, none of the issues is present, as we focus on assessing past performance (feeding back) and on feedback whose purpose is designed and motivated by the researchers’ interests.

* The authors thank an anonymous reviewer for suggesting these discussions.”

LIMITATIONS AND FUTURE RESEARCH:

“(…) the present study can be helpful for research on the reactions to feedback within the self-motives literature. The present study was developed with participants who were in a passive position in regard to the feedback process, i.e., the design and the purpose of feedback were based on the interests of an external party (the instructors/researchers). However, there is a whole additional literature on people who actively seek for, and react to, feedback due to self-enhancement and self-improvement interests (Anseel, Lievens, & Levy, 2007)*.

* The authors thank an anonymous reviewer for suggesting these discussions.”

----

REVIEWER #1

Concerning your methodology, I found it a creative approach to combine thought and lab experiments. However, the rationale for this combination should be further elaborated. The paper currently provides brief reasoning, but it could be more thoroughly explained—why precisely is the combination with thought experiments important?

AUTHORS

Thank you for asking us to develop this further. We added several new notes about the mixed experimental design, such as this one:

“On one hand, experiments are usually too limited to help understand human attitudes and behaviors, since any attitude or behavior is too complex to be explained by manipulating a few independent variables and observing their effects through also a limited chain of variables. On the other hand, it would be too complicated, if not impossible, to design and perform an experiment with reasonably many variables. Moreover, if the experiment also involves multiple dependent variables, it may become virtually impossible to either do it or develop satisfactorily trustful results. For these and other reasons, scholars have invested in thought experiments to shed light on possible causal chains of events that are too complex to submit to statistical tests whereas certain phenomena require the exploration of multiple causal links even if not currently possible in an empirical experiment.”

Moreover, we had originally planned to test how overconfident people react to feedback they receive about their performance in tasks. During the study, we realized that feedback was not independent of other concurring constructs in the formation of one’s reactions. For this reason, we decided to conduct an in-depth review of those constructs and elaborate on their manifestation during the experiment to help explain what we have found. Could we have excluded those events from our model and theoretical discussion? Possibly yes, but then we would have ignored reasonably influential phenomena that are helpful to explain our data as well as to inform future research.

----

REVIEWER #1

In the paper, you state: "We informed the participants about the experiment only after its conclusion, when we asked for their individual consent to analyze and publish the data." This implies that participants were unaware they were taking part in an experiment beforehand. Do you consider this ethically sound, did you coordinate this with the ethics committee of your university?

AUTHORS

Thank you for asking us about this issue, so that we can clarify it in the paper. We included an end note on this issue in a sentence of the Methods section, as follows:

“We informed the participants about the experiment only after its conclusion, when we asked for their individual consent to analyze and publish the data*.”

* “The statement on ethical research was shared with the publisher.”

In Brazil (where the study was done), we can inform the participants about an experiment after the collection of data, if the data collection process and the data do not pose any foreseeable risk to the participants. In our study, the experiment was part of a regular classroom exercise in a course, i.e., it was virtually undistinguishable if compared to other classroom meetings. One question may arise as per the student who left the room after the first task. She left the room after receiving negative feedback, but she actually performed low (5.25 out of 10 possible points in ATP) while expecting high performance (9.33 average CSE). She was then informed about the experiment (and that the grade would not be registered) as soon as she left the room. On a last note, we uploaded to Behavioral Sciences the individual consent form that each student filled for us to use their data.

----

REVIEWER #1

Additionally, was there a risk that participants from the first session informed those in later sessions about the nature of the experiment?

AUTHORS

Thanks, we had not considered this possibility. But we believe that it was not influential. We now included the following clarification note in the Limitations section:

“A fourth limitation* is that we had not considered the possibility that the students in the morning section could inform those in the evening section about the experiment. In the university where the study was done (much like everywhere), most students take classes independently and work in the remaining hours of the day, thus it is very unlikely that they would have reasons to immediately share information with others about the activities in the classroom. Also, we did not inform the first classroom that the same exercise would be done again with other students.

* The authors thank an anonymous reviewer for this remark.”

----

REVIEWER #1

You also mention in the paper that “This was done with a 10-minute video about that dilemma’s principles.” How did you verify that participants actually watched the video and understood the instructions? What was your manipulation check?

AUTHORS

The video was shown in the classroom with everyone present. The video had no instructions, but an illustration of the prisoners’ dilemma. We believe that we can assume that everyone paid attention to the video, since we would do a point-earning activity based on it in the following minutes. Anyway, in our original submission we had already anticipated a suggestion for future research regarding the priming procedure, as follows:

“(…) future research can test the effects of the use of priming to introduce a decision problem. In our study, priming was used to level the participants’ awareness about the tasks to be done, but it is possible that the model-observer similarity hypothesis and the task appropriateness hypothesis explain the cognitive gains and losses of some individuals due to personal identification with the priming stimulus or with those who produced it (Schunk, 1987; Hoogerheide et al., 2017, 2018).”

----

REVIEWER #1

Also the check for feedback: Has it been verified whether each student actually read the feedback? After all, it is not possible to determine which students genuinely reflected on the feedback and which did not.

AUTHORS

The feedback consisted of a very short message printed in a small paper given to each student personally by the researchers. The messages read as: “Thank you. Your performance was satisfactory.” (positive), “Thank you. Your performance was not satisfactory.” (negative) and “Thank you. Let’s begin the second task.” (neutral). We asked each student to read the message as part of the protocol. We believe that it is reasonable to assume that by looking at the paper each student has read the message.

----

REVIEWER #1

How did the authors determine the threshold of a 2.5-point (or more) difference between AvCSE and ATP for classifying an individual as overconfident?

AUTHORS

As explained before, we do not know of any reference to identify overconfident individuals, thus we developed a pragmatic heuristic that is described in the following paragraph:

“We are not aware of any method to identify overconfident individuals, thus we relied on experience and on an empirically based, post-hoc procedure: we identified individuals with higher AvCSE than their ATP and selected the individuals with differences between AvCSE and ATP equal or superior to 2.5 points in the scale. Such a magnitude takes into consideration several issues, like the arithmetic difference itself (a very prudent, arguably expressive 25% difference between expectation and effectiveness) and the accuracy, or lack of, with which one is able to assess and report three different types of CSE from which AvCSE is computed. Moreover, it is not any level of CSE that exceeds ATP that should be considered an indication of overconfidence, since moderate-high levels of CSE have been reported as desirable for college students to achieve technology use effectiveness (Porto-Bellini et al., 2016). Two additional reasons support our heuristic: first, by examining Table 3, one realizes that 2.5 is well above the other AvCSE-ATP differences, except for only six measures between 2.1 and 2.3 reported in the two tasks (out of 108 differences computed in total), meaning that 2.5 is significantly above 94% of all AvCSE-ATP differences; and the second reason is that a range of 2.5 is also well above the difference needed for a student to move between grades in the Grade Point Average (GPA) system. Therefore, 2.5 and above can be reasonably accepted as a manifestation of overconfidence. On a final note, such a rationale does not mean that difference scores below 2.5 may not represent overconfidence too. However, since this study is not a statistical one, there are no implications for not considering the below-2.5 individuals in the analysis.”

----

REVIEWER #1

Furthermore, you write that “We developed three five-item scales based on Compeau and Higgins (1995), Murphy et al. (1989), and Gist, Schwoerer, and Rosen (1989), with scores ranging from 0 to 10 to measure the three types of CSE we originally proposed.” How were these scales developed? More information is needed on this point, possibly in a supplemental section.

AUTHORS

Thank you for asking us to clarify this issue, as it allows us to highlight some contributions we make to previous CSE scales. We have now expanded the explanation about scale development as follows:

“We developed three five-item scales following patterns found in data collection instruments since the first CSE studies, such as those in Compeau and Higgins (1995), Murphy et al. (1989) and Gist, Schwoerer, and Rosen (1989). Those instruments typically consist of short statements about a specific situation of computer use, sometimes referring to a specific tool as well. We replicated this pattern. However, there are two fundamental differences between our scale and numerous others. First, while some instruments (such as those three mentioned above) employ scales of confidence to measure self-efficacy, we did not find any rationale for using confidence instead of self-efficacy ratings—which are different constructs (Morony et al., 2013). Bandura and Cervone (1986) in fact discuss how capable, not how confident, one is. Therefore, we asked participants to rate how capable they think they are to accomplish certain actions on the computer. And a second difference between our scale and extant ones is that ours ranges from 0 to 10 (therefore, it has a neutral point), not from 1 to 10 (with no neutral point). We did not find any argument in favor of not having the neutral point. On the contrary, Compeau and Higgins (1995, p. 201-211) mention exactly point number “5” in their scale as representing “moderately confident” individuals, thus suggesting that they see number “5” as a neutral point; however, point number “5” does not stand exactly in the middle of their scale. Our instrument is available in Appendix A. It includes three sections, one for each of the CSE types we developed based on the rationale presented in the literature review. The scale was conceptually discussed with our research group in numerous rounds as well as in a version presented at the Americas Conference on Information Systems (reference not available before peer review).”

In the Appendix, we also added the following note:

“The following questionnaire must be adapted to each specific computer tool and decision problem.”

----

REVIEWER #1

Next, you explain that “The independent variable was homogeneously administered to individuals according to the feedback groups (GPOS, GNEG, or GCTRL) and irrespective of actual individual performance.” This should be made clearer—does this mean that even high-performing students (who felt they had performed well) could receive negative feedback? Could this not be a confounding factor affecting the study’s results?

AUTHORS

Yes, you are correct. We distributed the same amount of positive, negative, and neutral feedback, i.e., 18 of each type (three types of feedback to a total of 54 participants). That is why we had three groups of people in the experiment: 18 individuals who received positive feedback, 18 individuals who received negative feedback, and 18 individuals who received neutral feedback. This also means that some individuals received “correct” feedback (positive feedback to good performance, and negative feedback to poor performance), some individuals received “incorrect” feedback (positive feedback to poor performance, and negative feedback to good performance), and some individuals did not receive any real feedback (the neutral feedback was just a pause message in between the two tasks). However, we added a note on why possible discrepancies between the type of feedback and actual performance are not expected to have misguided the students, as follows (special attention to the three last sentences):

“Feedback was given individually in written form to each student after completing Task #1. Three feedback groups were formed (much like the affect manipulation of feedback given to students in Cabestrero et al., 2018): individuals who received positively stated feedback (“Thank you. Your performance was satisfactory.”), individuals who received negatively stated feedback (“Thank you. Your performance was not satisfactory.”), and individuals who received neutral feedback (“Thank you. Let’s begin the second task.”). In selecting the individuals to receive each type of feedback, the instructors analyzed each student’s ATP for Task #1 and then equally distributed each type of feedback (positive, negative, and neutral) to those with actually satisfactory and not satisfactory performance, as follows (Figure 2): half of each type of feedback (positive, negative, and neutral) was randomly given to each of the two performance groups (students with satisfactory and not satisfactory ATP). The criterion for deciding whether a performance was satisfactory or not was the official score to pass an exam in the university (at least 7 points out of a maximum of 10 possible points), but the instructors did not explain the reason for either the positive or the negative feedback. As such, the participants were free to reflect on all possible performance outcomes (about the general use of the computer, the use of the specific computer tool, or the decision-making activity). As shown later (Table 3), no student achieved the minimum or the maximum scores possible, thus all feedback messages could be accepted as meaningful.”

----

REVIEWER #1

Sidenote: There are clear differences between the groups Gctrl, Gpos, and Gneg (e.g., gender, internship experience, etc.). Could these differences have influenced the results? Can you clarify this more?

AUTHORS

Many studies do not discuss at a minimum their demographic data. Here is an example from a study recommended for our revision, where the authors state that “individual differences were not taken into account as possible moderator variables of the FFI’s effect on job performance” (Budworth, Latham, & Manroop, 2015, p. 51). Anyway, we have deployed all acceptable means to have comparable groups in terms of the most important demographic data that we were granted access to by the ethical guidelines. We discuss those data in the paragraph below and in its associated Table 1:

“Participants were undergraduate students of Business Administration assigned to two different classrooms (morning and evening sections) of a course called Administrative Informatics. A total of 54 students participated in the study, 29 from the morning section and 25 from the evening section. We assigned them evenly and randomly to three groups: 18 students to an experimental group that would receive positive feedback on task performance (GPOS), 18 students to an experimental group that would receive negative feedback (GNEG), and 18 students to a control group that would receive neutral feedback (GCTRL). Besides the equal number of students in each group, their assignment to the groups was partially controlled for pairing, i.e., the demographic profile of each group was partially homogenized regarding the most typical variance-generating variables as judged from the historical demographic distribution in similar classrooms (Table 1). Full pairing of all variables was evidently not an option, since pairing the most influential variables impedes full pairing of other, less influential ones. For instance, one may question why gender was not a priority for pairing. The reason is, no interactions were found between gender and intelligence theories on overconfidence in Ehrlinger, Mitchum, and Dweck’s (2016) study on overconfidence with undergraduate students. Broadly, those authors found a lack of scholarly knowledge regarding the demographic aspects of overconfident individuals, rather assuming that views of intelligence (entity/fixed or incremental/malleable) and the locus of attention (difficult or easy tasks) may explain self-assessments of performance and effects on learning. On a note of caution, Biernat and Danaher (2012) found differences in immediate reactions to subjective interpretations of feedback according to gender and race, but their study is not comparable to ours in many ways (e.g., their experimental tasks involved leadership roles, and their focal measurement was the level of importance the participants assigned to those roles); and Narciss et al.’s (2014) study found gender differences in learning performance of students under tutoring schemes, which is also not the case in our study.”

Mentioned source:

Budworth, M.-H., Latham, G. P., & Manroop, L. (2015). Looking forward to performance improvement: A field test of the feedforward interview for performance management. Human Resource Management, 54(1), 45-54, https://doi.org/10.1002/hrm.21618

----

REVIEWER #1

You also mention this: “Therefore, this may explain that student’s behavior as well as why other students had difficulty adjusting their CSE levels. Since that student was a woman, another possible explanation, according to a review on self-perceptions by Chevrier et al. (2020), relates to female college students being more self-critical about their academic abilities and more affected by self-esteem issues that address their intellectual abilities, scholastic competence, and expectations about college life.” This explanation seems somewhat reductionist—it is based on a single case and does not seem entirely relevant to the study’s broader scope, particularly since gender is stated elsewhere as not being a relevant variable.

AUTHORS

Thank you indeed for this remark, which allows us to reflect on this passage. We concluded that there is nothing wrong with it, for two reasons: first, if we exclude it, some readers (or another reviewer) may feel the need for some comment on the fact that one student left the experiment, what is easily perceived when reading Table 3; and second, we believe that the Reviewer has used an inverse logic: we are not developing a theory based on a single case, but using a theory to explain a single case. Anyway, we are open to discuss this further if this is a critical issue.

----

REVIEWER #1

Discussion part: However, the most significant issue with the study, in my view, is the sample size. The authors acknowledge this in the discussion: “Therefore, all we can do before better knowledge is available in a given situation (either with or without statistically sound measures) is to develop a temporarily satisfactory explanation based on our own factual experience and judgment about how and why the events of our interest manifest as they do. In the present study, we have opted to invest our best efforts in analytical reasoning rather than having our conclusions constrained within the statistically accepted boundaries.” A stronger justification is required—perhaps by referencing more other studies that worked with similarly small samples or by drawing on relevant literature to reinforce the methodological approach. A more robust argument (either methodological or literature-based) is necessary to justify the small sample size.

AUTHORS

We tried to provide the needed rationale for why our sample is acceptable when considered the epistemological stance of our study. Anyway, now we added several other clarification notes. We also mention one study that employs analytical reasoning (even if the authors do not use this specific term) to analyze data from an ex-post-facto study (Porto-Bellini, Pereira, & Becker, 2020).

Two issues can be considered. First, our sample is even larger (54 vs. 46) than that of a comparable study published in a reputed journal (Biernat & Danaher, 2012). However, contrary to doing statistical tests like they did, we do not think such sample sizes are appropriate for statistical manipulation. Therefore, we coherently did not engage in statistical tests. The second issue refers to how we addressed the analysis of data. This is explained in several parts of the paper. If we had done a qualitative observation of 54 individuals doing tasks on the computer, we would probably exceed what is needed to develop insightful knowledge. However, since we mention that we did an experiment, readers are inclined to judge it from the perspective of statistical methods. But experiments do not imply statistical manipulation. Rather, an experiment refers to the design of the variables, their relations, and the administration of treatments. Experiments exist much before statistics. In the present study, we collected rich data from a rigorously designed experiment, with deep personal involvement with the context and with the participants (we were instructors of those students) and the data have ecological validity for referring to an actual classroom situation. Since we do not agree to doing statistical manipulations with only 54 sources of data, we then resorted to a qualitative analysis of the data. It is our view that our data should not be discussed with the same criteria used for regular experiments.

We added in the text a few new notes to support our analytical approach, among them the following two paragraphs:

“Due to the presence of some conjectures in the experimental design and the number of participants (54 in total), the hypotheses stated before were not submitted to statistical tests. Rather, they were handled with analytical reasoning (explained later). Such a decision was aimed at not infringing statistical assumptions that are otherwise expected for construct validation. In contrast, our experiment is of an exploratory, interpretive nature and highly dependent on the phenomenological experience of the researchers with the experimental context and the participants. Indeed, the perfect balance between internal validity (consistency) and external validity (generalization) is always a challenge. Our study prioritizes internal validity in many senses (it includes an arguably true field experiment with high ecological validity and immersion of the researchers in the context and with the participants) while lacking evidence for statistical validity. Other studies do the exact opposite when using an experimental design. For instance, in a study by Budworth, Latham, and Manroop (2015, p. 52) on feedforward interventions, ‘internal validity was decreased in favor of external validity’ and ‘random assignment of participants to a control group […] was not possible’.”

“The data analysis procedures are characterized as analytical reasoning, which is a helpful approach in complex situations or in the lack of statistically significant data. It is not a particular method, rather being a moniker used in practice to refer to an ample set of analytical skills that include the insightful, ad-hoc elaboration of procedures to solve a pragmatic problem. One example in the IT literature is available in Porto-Bellini, Pereira, and Becker’s (2020) ex-post-facto study on the cognitive and behavioral archetype of software developers that correlate with the success of enterprise systems projects. While our study’s sample (54 subjects) is numerically comparable to others (e.g., 46 subjects in Biernat & Danaher, 2012), it is not large enough to make us comfortable to perform statistically based comparisons of means. Also, since we coherently did not collect objective data for the thought experiment parts of the experimental design, submitting the available data to statistical tests would not help address those parts too. In contrast, by resorting to analytical reasoning, we invested our best analytical efforts to understand and explain the patterns and idiosyncrasies in the data, which are highly dependent on the phenomenological experience of the researchers with the application context and the technology users. Analytical reasoning is in fact expected to produce deep understanding of a situation that would otherwise be constrained within the limits of cold statistical manipulation. For this reason, it is used in U.S. universities to build a more complete set of analytical skills among students and employees. Moreover, the present study was done in a highly idiosyncratic society—the Brazilian northeastern region—thus demanding immersive experience of the researchers to discuss certain attitudinal and behavioral patterns that are possibly present in the data.”

Mentioned sources:

Biernat, M., & Danaher, K. (2012). Interpreting and reacting to feedback in stereotype-relevant performance domains. Journal of Experimental Social Psychology, 48(1), 271-276, https://doi.org/10.1016/j.jesp.2011.08.010

Budworth, M.-H., Latham, G. P., & Manroop, L. (2015). Looking forward to performance improvement: A field test of the feedforward interview for performance management. Human Resource Management, 54(1), 45-54, https://doi.org/10.1002/hrm.21618

Porto-Bellini, C. G., Pereira, R. C. F., & Becker, J. L. (2020). Emergent customer team performance and effectiveness: An ex-post-facto study on cognition and behavior in enterprise systems implementation. Communications of the AIS, 47, 550-582, http://dx.doi.org/10.17705/1CAIS.04726

----

REVIEWER #1

Differences are also observed in the control group, both in ATP and AvCSE. How can these differences be explained? This is currently not considered in the conclusions.

AUTHORS

We believe to have addressed this issue when we discussed the possible influence of learning on the task, as both experimental tasks shared the same underlying logic. However, we took the Reviewer’s comment as a motivation to revise and improve the discussion of data, which we believe to have done dutifully in this revised version.

----

REVIEWER #1

The practical implications could be more specifically articulated. As they stand, they remain somewhat vague in terms of what the study offers in concrete terms. The utility of the final implication (the tool) is also not entirely clear.

AUTHORS

Thank you indeed for sharing this concern. We reflected extensively about it but did not find many ways to significantly improve this section. We added some clarification notes, but our key practical contributions remain the same, as follows in this newly added sentence:

“…we provide three new scales for CSE (Appendix A), two versions of a scenario-based computer-aided decision task for use in the classroom and in professional training (Appendix B), an experimental design, and numerous insights into the significant numbers of overconfident individuals that may be present in a work group (since we found around one-third of overconfident individuals in our sample, using a very conservative measure)”

----

REVIEWER #1

In the final conclusion, the authors state: “This study aimed at understanding the reaction of overconfident students to feedback received from their instructors on task performance under the premises of technology use effectiveness.” This appears to be a rather narrow contribution—what is the broader significance of understanding this? The value of the paper likely extends beyond this and should be more explicitly emphasized.

AUTHORS

In our scholarly fields, we are used to summarizing the conclusions in just a few sentences. Moreover, the text is already long (more than 16,000 words) and the theoretical and practical implications were discussed in a specific section. We now improved a few reading issues but maintained the conclusions in summarized form.

----

FINAL COMMENTS:

We believe that the revised manuscript offers the due answers to the Reviewers’ concerns as well as an improved reading experience to the prospective reader of Behavioral Sciences. We tried to satisfactorily address each review comment, and we also implemented a few additional improvements by ourselves.

We thank again the Reviewer for the opportunity granted to us to revise our study and the manuscript, and we remain open to discuss further issues if needed.

Sincerely,

The authors

Reviewer 2 Report

Comments and Suggestions for Authors

Dear authors. Please find my review of your article bellow.

Title: Computer Self-efficacy and Reactions to Feedback in the Class- 2 room: Reopening the Debate in an Interpretive Experiment with Overconfident Students

Brief summary

This study explores how feedback influences computer self-efficacy (CSE) and task performance among students, with a focus on those exhibiting overconfidence. By conducting an experiment with 54 undergraduate students, the research examines the effects of different types of feedback—neutral, positive, and negative—on students’ self-perceptions and actual performance. The findings contribute to the discussion on feedback’s role in learning and introduce a new three-dimensional model of CSE.

My general comments are:

Strengths of your article:

  • Your study investigates an important topic within education and technology.
  • Mixed-methods approach provides depth to the findings your research.
  • The proposed three-dimensional model of CSE is an innovative contribution.
  • The literature review is thorough and includes recent research.
  • The study has practical implications for educational and workplace settings.

Areas where improvement is suggested and needed:

1. Hypothesis development: The study would benefit from clearly stated hypotheses. Even though the research is exploratory, well-defined hypotheses would strengthen its structure.

2. Methodological concerns:

    • Small sample size: The study relies on data from only 54 students, which limits the ability to generalize findings to a broader population. A larger and more diverse sample would strengthen the validity of the conclusions.
    • Feedback assignment bias: The process of assigning feedback does not seem to account for differences in students’ baseline performance. As a result, some students may have received feedback that did not accurately reflect their actual capabilities, potentially influencing their reactions in unintended ways.
    • Complexity of mixed experimental design: The study incorporates both thought experiments and lab-based experiments, which, while innovative, make replication challenging. Thought experiments depend on theoretical reasoning rather than empirical measurement, and combining them with lab experiments introduces variability that future researchers may struggle to reproduce under the same conditions.

3. Feedback interpretation:

  • It is unclear if participants viewed the feedback as applying strictly to their computer skills or to their decision-making abilities more broadly.
  • More qualitative insights on how students perceived the feedback would enhance the findings.

4. Statistical analysis:

  • The study presents descriptive trends, but inferential statistics (e.g., ANOVA, regression) could better support its conclusions.
  • A more robust statistical approach would improve the credibility of the findings.

Specific comments

  • Lines 56-77: The connection between technology use effectiveness and feedback mechanisms should be more explicitly defined.
  • Table 1: The rationale for demographic controls should be expanded to clarify their impact on results.
  • Figures 3 & 4: While helpful, additional explanation regarding variance between groups would improve understanding.
  • Ethical considerations: The methods section should specify whether ethical approval was obtained and how informed consent was handled.
  • Limitations section: While some limitations are discussed, potential response biases (e.g., social desirability) should be more thoroughly examined.

The study provides an interesting perspective on feedback and self-efficacy, but improvements in research design, statistical analysis, and methodological transparency are necessary. Addressing these concerns will enhance the paper’s contribution to the field.

Hope you will find this comments valuable to improve your research article.

Regards.

Reviewer

Author Response

(If needed, please see also attached the responses given to the other Reviewer.)

Dear Reviewer:

We write this revision letter regarding our manuscript code “behavsci-3473350” (originally titled “Computer Self-efficacy and Reactions to Feedback in the Classroom: Reopening the Debate in an Interpretive Experiment with Overconfident Students”). We are pleased to know that the Editor and the Reviewers found merits in the manuscript and see a possibly good future for it after a revision.

We are always happy to receive reviews as they make it perfectly clear how important the peer review process is and how true it is that multiple and acceptable perspectives exist about any work. The two reviews we received showed to us the need to improve our communication to the reader in many aspects, as well as that our work is limited by additional factors besides those we had initially identified. We thus carefully revisited the paper in full and paid particular attention to addressing the review comments both in the revised paper and in this response letter. Here, we discuss each review comment as well as a few additional opportunities for improvement that we identified and addressed during the revision.

Overall, the Reviewers found issues in need of improvement in terms of the study’s conceptual and methodological aspects. While the timeframe for this revision is a tight one (only 10 days), we believe to have solved most issues by better positioning our study within the epistemological spectrum and communicating it properly to the reader. In the next pages, we describe in due detail the Reviewers’ comments and how we addressed them in the revised paper. We have also uploaded, to the manuscript handling system, a file with all revision marks.

---

REVIEWER #2

Brief summary:

This study explores how feedback influences computer self-efficacy (CSE) and task performance among students, with a focus on those exhibiting overconfidence. By conducting an experiment with 54 undergraduate students, the research examines the effects of different types of feedback—neutral, positive, and negative—on students’ self-perceptions and actual performance. The findings contribute to the discussion on feedback’s role in learning and introduce a new three-dimensional model of CSE.

Strengths of your article:

Your study investigates an important topic within education and technology. Mixed-methods approach provides depth to the findings your research. The proposed three-dimensional model of CSE is an innovative contribution. The literature review is thorough and includes recent research. The study has practical implications for educational and workplace settings.

AUTHORS

We thank the Reviewer for acknowledging the intentions we had with this study.

---

REVIEWER #2

Areas where improvement is suggested and needed:

  1. Hypothesis development: The study would benefit from clearly stated hypotheses. Even though the research is exploratory, well-defined hypotheses would strengthen its structure.

AUTHORS

This is a challenging demand for us. We are extremely attached to formalism, and as such we are not comfortable with stating hypotheses if the study does not include statistical manipulations. Anyway, we now added six explicit hypotheses along with notes that they were not tested statistically.

---

REVIEWER #2

  1. Methodological concerns:

Small sample size: The study relies on data from only 54 students, which limits the ability to generalize findings to a broader population. A larger and more diverse sample would strengthen the validity of the conclusions.

AUTHORS

Thank you for the opportunity to explain this further. Two issues can be considered. First, our sample is even larger (54 vs. 46) than that of a comparable study published in a reputed journal (Biernat & Danaher, 2012). However, contrary to doing statistical tests like they did, we do not think such sample sizes are appropriate for statistical manipulation. Therefore, we coherently did not engage in statistical tests. The second issue refers to how we addressed the analysis of data. This is explained in several parts of the paper. If we had done a qualitative observation of 54 individuals doing tasks on the computer, we would probably exceed what is needed to develop insightful knowledge. However, since we mention that we did an experiment, readers are inclined to judge it from the perspective of statistical methods. But experiments do not imply statistical manipulation. Rather, an experiment refers to the design of the variables, their relations, and the administration of treatments. Experiments exist much before statistics. In the present study, we collected rich data from a rigorously designed experiment, with deep personal involvement with the context and with the participants (we were instructors of those students) and the data have ecological validity for referring to an actual classroom situation. Since we do not agree to doing statistical manipulations with only 54 sources of data, we then resorted to a qualitative analysis of the data. It is our view that our data should not be discussed with the same criteria used for regular experiments.

We added in the text a few new notes to support our analytical approach, among them the following two paragraphs:

“Due to the presence of some conjectures in the experimental design and the number of participants (54 in total), the hypotheses stated before were not submitted to statistical tests. Rather, they were handled with analytical reasoning (explained later). Such a decision was aimed at not infringing statistical assumptions that are otherwise expected for construct validation. In contrast, our experiment is of an exploratory, interpretive nature and highly dependent on the phenomenological experience of the researchers with the experimental context and the participants. Indeed, the perfect balance between internal validity (consistency) and external validity (generalization) is always a challenge. Our study prioritizes internal validity in many senses (it includes an arguably true field experiment with high ecological validity and immersion of the researchers in the context and with the participants) while lacking evidence for statistical validity. Other studies do the exact opposite when using an experimental design. For instance, in a study by Budworth, Latham, and Manroop (2015, p. 52) on feedforward interventions, ‘internal validity was decreased in favor of external validity’ and ‘random assignment of participants to a control group […] was not possible’.”

“The data analysis procedures are characterized as analytical reasoning, which is a helpful approach in complex situations or in the lack of statistically significant data. It is not a particular method, rather being a moniker used in practice to refer to an ample set of analytical skills that include the insightful, ad-hoc elaboration of procedures to solve a pragmatic problem. One example in the IT literature is available in Porto-Bellini, Pereira, and Becker’s (2020) ex-post-facto study on the cognitive and behavioral archetype of software developers that correlate with the success of enterprise systems projects. While our study’s sample (54 subjects) is numerically comparable to others (e.g., 46 subjects in Biernat & Danaher, 2012), it is not large enough to make us comfortable to perform statistically based comparisons of means. Also, since we coherently did not collect objective data for the thought experiment parts of the experimental design, submitting the available data to statistical tests would not help address those parts too. In contrast, by resorting to analytical reasoning, we invested our best analytical efforts to understand and explain the patterns and idiosyncrasies in the data, which are highly dependent on the phenomenological experience of the researchers with the application context and the technology users. Analytical reasoning is in fact expected to produce deep understanding of a situation that would otherwise be constrained within the limits of cold statistical manipulation. For this reason, it is used in U.S. universities to build a more complete set of analytical skills among students and employees. Moreover, the present study was done in a highly idiosyncratic society—the Brazilian northeastern region—thus demanding immersive experience of the researchers to discuss certain attitudinal and behavioral patterns that are possibly present in the data.”

Mentioned sources:

Biernat, M., & Danaher, K. (2012). Interpreting and reacting to feedback in stereotype-relevant performance domains. Journal of Experimental Social Psychology, 48(1), 271-276, https://doi.org/10.1016/j.jesp.2011.08.010

Budworth, M.-H., Latham, G. P., & Manroop, L. (2015). Looking forward to performance improvement: A field test of the feedforward interview for performance management. Human Resource Management, 54(1), 45-54, https://doi.org/10.1002/hrm.21618

---

REVIEWER #2

Feedback assignment bias: The process of assigning feedback does not seem to account for differences in students’ baseline performance. As a result, some students may have received feedback that did not accurately reflect their actual capabilities, potentially influencing their reactions in unintended ways.

AUTHORS

Yes, you are correct. We distributed the same amount of positive, negative, and neutral feedback, i.e., 18 of each type (three types of feedback to a total of 54 participants). That is why we had three groups of people in the experiment: 18 individuals who received positive feedback, 18 individuals who received negative feedback, and 18 individuals who received neutral feedback. This also means that some individuals received “correct” feedback (positive feedback to good performance, and negative feedback to poor performance), some individuals received “incorrect” feedback (positive feedback to poor performance, and negative feedback to good performance), and some individuals did not receive any real feedback (the neutral feedback was just a pause message in between the two tasks). However, we added a note on why possible discrepancies between the type of feedback and actual performance are not expected to have misguided the students, as follows (special attention to the last three sentences):

“Feedback was given individually in written form to each student after completing Task #1. Three feedback groups were formed (much like the affect manipulation of feedback given to students in Cabestrero et al., 2018): individuals who received positively stated feedback (“Thank you. Your performance was satisfactory.”), individuals who received negatively stated feedback (“Thank you. Your performance was not satisfactory.”), and individuals who received neutral feedback (“Thank you. Let’s begin the second task.”). In selecting the individuals to receive each type of feedback, the instructors analyzed each student’s ATP for Task #1 and then equally distributed each type of feedback (positive, negative, and neutral) to those with actually satisfactory and not satisfactory performance, as follows (Figure 2): half of each type of feedback (positive, negative, and neutral) was randomly given to each of the two performance groups (students with satisfactory and not satisfactory ATP). The criterion for deciding whether a performance was satisfactory or not was the official score to pass an exam in the university (at least 7 points out of a maximum of 10 possible points), but the instructors did not explain the reason for either the positive or the negative feedback. As such, the participants were free to reflect on all possible performance outcomes (about the general use of the computer, the use of the specific computer tool, or the decision-making activity). As shown later (Table 3), no student achieved the minimum or the maximum scores possible, thus all feedback messages could be accepted as meaningful.”

---

REVIEWER #2

Complexity of mixed experimental design: The study incorporates both thought experiments and lab-based experiments, which, while innovative, make replication challenging. Thought experiments depend on theoretical reasoning rather than empirical measurement, and combining them with lab experiments introduces variability that future researchers may struggle to reproduce under the same conditions.

AUTHORS

Thank you for asking us to develop this further. Generalization and replication of data were not our intention. Rather, we envisioned the elaboration of an experimental design that can be replicated and a phenomenological understanding of attitudes and behaviors in a specific sample of students. We added several new notes to explain our stance, such as the following ones:

“On one hand, experiments are usually too limited to help understand human attitudes and behaviors, since any attitude or behavior is too complex to be explained by manipulating a few independent variables and observing their effects through also a limited chain of variables. On the other hand, it would be too complicated, if not impossible, to design and perform an experiment with reasonably many variables. Moreover, if the experiment also involves multiple dependent variables, it may become virtually impossible to either do it or develop satisfactorily trustful results. For these and other reasons, scholars have invested in thought experiments to shed light on possible causal chains of events that are too complex to submit to statistical tests whereas certain phenomena require the exploration of multiple causal links even if not currently possible in an empirical experiment.”

“… the present study was done in a highly idiosyncratic society—the Brazilian northeastern region—thus demanding immersive experience of the researchers to discuss certain attitudinal and behavioral patterns that are possibly present in the data. On a final note, generalization and replication of data were not the intention of this study. Rather, it envisioned the elaboration of an experimental design that can be replicated and a phenomenological understanding of attitudes and behaviors in a specific sample of students.”

We had originally planned to test how overconfident people react to feedback they receive about their performance in tasks. During the study, we realized that feedback was not independent of other concurring constructs in the formation of one’s reactions. For this reason, we decided to conduct an in-depth review of those constructs and elaborate on their manifestation during the experiment to help explain what we have found. Could we have excluded those events from our model and theoretical discussion? Possibly yes, but then we would have ignored reasonably influential phenomena that are helpful to explain our data as well as to inform future research.

---

REVIEWER #2

  1. Feedback interpretation: It is unclear if participants viewed the feedback as applying strictly to their computer skills or to their decision-making abilities more broadly. More qualitative insights on how students perceived the feedback would enhance the findings.

AUTHORS

Good point. Indeed, we did not manipulate any of the specific types of CSE. While this may be a limitation, it is coherent with the fact that we focused on average CSE (AvCSE) when comparing the CSE levels to actual task performance (ATP). It is precisely because we did not specify the type of CSE in the feedback message that most students might have accepted their feedback as a real one, as we describe in this passage about the feedback plan:

“… the instructors did not explain the reason for either the positive or the negative feedback. As such, the participants were free to reflect on all possible performance outcomes (about the general use of the computer, the use of the specific computer tool, or the decision-making activity).”

Nevertheless, we now included this issue in the Limitations section, as follows:

“… we did not manipulate any of the specific CSE types (G-CSE, T-CSE, P-CSE). While this may be a limitation, it is coherent with the fact that we focused on average CSE (AvCSE) when comparing the CSE levels to actual task performance (ATP).”

---

REVIEWER #2

  1. Statistical analysis: The study presents descriptive trends, but inferential statistics (e.g., ANOVA, regression) could better support its conclusions. A more robust statistical approach would improve the credibility of the findings.

AUTHORS

We do agree that statistical manipulations would provide objective support for the findings. However, as commented before, our sample is not sufficient for such manipulations, regardless of other studies doing so with smaller samples. But most importantly, also as commented before, more “qualitative” studies like ours sometimes allow for deeper analyses based on the real experience of researchers with the research context and subjects, what we firmly believe is the case in our study.

---

REVIEWER #2

Specific comments Lines 56-77: The connection between technology use effectiveness and feedback mechanisms should be more explicitly defined.

AUTHORS

Thank you for pointing this out. We rewrote several lines and the paragraphs now read as follows:

“We designed a study on how technology users perceive themselves and react to external perceptions about themselves (from feedback provided by an authorized agent) regarding their actual performance in computer-aided tasks. We aimed to estimate how mindful one is about performing tasks on the computer and how he or she processes others’ assessments about his or her performance and correspondingly adjusts attitudes and behaviors. In doing so, we aim to add to the emergent stream of research on technology use effectiveness (Burton-Jones & Grange, 2013; Porto-Bellini, 2018), which in some sense improves the dominant models of technology use by framing not only what motivates use, but also what makes one achieve arbitrarily defined use purposes by understanding what needs to be done and the means to doing it.

Moreover, we pay special attention to the presence of overconfidence in individuals performing computer-aided tasks, which is a particularly important trait in the study of human cognition and decision making (Kahneman & Klein, 2009). Overconfidence is here framed as a post-hoc measure of unrealistically high self-efficacy beliefs, what represents a cognitive limitation towards one’s effectiveness in using the digital technologies. We thus searched for an answer to the question of whether feedback on performance, as well as the valence of feedback, has any impact on overconfident individuals in their computer self-efficacy (CSE) beliefs and actual task performance. To answer it, we designed an experiment with undergraduate students of Business Administration who were regularly assigned to perform computer-aided decision tasks. We submitted the participants to two sequential, similar computer-aided decision tasks and measured three newly defined dimensions of CSE beliefs before each task (general CSE, problem-related CSE, and tool-related CSE); and, after each task, we measured their self-perceptions of performance as well as their actual performance. In between the two tasks, we provided each participant with one of three types of manipulated feedback: neutral, negative, and positive feedback on their task performance. An important characteristic of the experiment is that it involved a realistic computer-aided business decision under real assessment by an authorized agent (the course’s instructor), instead of any computer use. With such an experimental design, we studied technology use with the theoretical lenses of use effectiveness, which requires the definition of the purposes of use and assumes that more than one perspective of use effectiveness may exist (in the present study, the perspective of students and the perspective of their instructor).”

---

REVIEWER #2

Table 1: The rationale for demographic controls should be expanded to clarify their impact on results.

AUTHORS

The demographic variables were considered for only two reasons: (1) to assure that the randomized assignment of individuals to the three experimental groups did not introduce any significant imbalance in the most critical demographic aspects, and to (2) possibly help analyze individual attitudes and behaviors. Since we had not handled the data with usual statistical procedures, the statistical concept of control variables does not apply to our analyses.

Many studies do not discuss at a minimum their demographic data. Here is an example from a study we reviewed, where the authors state that “individual differences were not taken into account as possible moderator variables of the FFI’s effect on job performance” (Budworth, Latham, & Manroop, 2015, p. 51). Anyway, we have deployed all acceptable means to have comparable groups in terms of the most important demographic data that we were granted access to by the ethical guidelines. We discuss those data in the paragraph below and in its associated Table 1:

“Participants were undergraduate students of Business Administration assigned to two different classrooms (morning and evening sections) of a course called Administrative Informatics. A total of 54 students participated in the study, 29 from the morning section and 25 from the evening section. We assigned them evenly and randomly to three groups: 18 students to an experimental group that would receive positive feedback on task performance (GPOS), 18 students to an experimental group that would receive negative feedback (GNEG), and 18 students to a control group that would receive neutral feedback (GCTRL). Besides the equal number of students in each group, their assignment to the groups was partially controlled for pairing, i.e., the demographic profile of each group was partially homogenized regarding the most typical variance-generating variables as judged from the historical demographic distribution in similar classrooms (Table 1). Full pairing of all variables was evidently not an option, since pairing the most influential variables impedes full pairing of other, less influential ones. For instance, one may question why gender was not a priority for pairing. The reason is, no interactions were found between gender and intelligence theories on overconfidence in Ehrlinger, Mitchum, and Dweck’s (2016) study on overconfidence with undergraduate students. Broadly, those authors found a lack of scholarly knowledge regarding the demographic aspects of overconfident individuals, rather assuming that views of intelligence (entity/fixed or incremental/malleable) and the locus of attention (difficult or easy tasks) may explain self-assessments of performance and effects on learning. On a note of caution, Biernat and Danaher (2012) found differences in immediate reactions to subjective interpretations of feedback according to gender and race, but their study is not comparable to ours in many ways (e.g., their experimental tasks involved leadership roles, and their focal measurement was the level of importance the participants assigned to those roles); and Narciss et al.’s (2014) study found gender differences in learning performance of students under tutoring schemes, which is also not the case in our study.”

Mentioned source:

Budworth, M.-H., Latham, G. P., & Manroop, L. (2015). Looking forward to performance improvement: A field test of the feedforward interview for performance management. Human Resource Management, 54(1), 45-54, https://doi.org/10.1002/hrm.21618 

---

REVIEWER #2

Figures 3 & 4: While helpful, additional explanation regarding variance between groups would improve understanding.

AUTHORS

Those figures, along with Table 3, were extensively revised as they convey the most important data we could gather from the participants. The Reviewer will see in the revised document that we clarified a few issues in the discussion of those figures, but we did not find ways to significantly improve the specific discussion requested by the Reviewer. We think that the request demands statistical analysis of variance, but as commented before we had not done it for methodological and epistemological reasons. We nevertheless discussed a few aspects of variance between groups both in the analysis of the raw data and the discussion of findings.

---

REVIEWER #2

Ethical considerations: The methods section should specify whether ethical approval was obtained and how informed consent was handled.

AUTHORS

Thank you for asking us about this issue, so that we can clarify it in the paper. We included an end note on this issue in a sentence of the Methods section, as follows:

“We informed the participants about the experiment only after its conclusion, when we asked for their individual consent to analyze and publish the data*.”

* “The statement on ethical research was shared with the publisher.”

In Brazil (where the study was done), we can inform the participants about an experiment after the collection of data, if the data collection process and the data do not pose any foreseeable risk to the participants. In our study, the experiment was part of a regular classroom exercise in a course, i.e., it was virtually undistinguishable if compared to other classroom meetings. One question may arise as per the student who left the room after the first task. She left the room after receiving negative feedback, but she actually performed low (5.25 out of 10 possible points in ATP) while expecting high performance (9.33 average CSE). She was then informed about the experiment (and that the grade would not be registered) as soon as she left the room. On a last note, we uploaded to Behavioral Sciences the individual consent form that each student filled for us to use their data.

---

REVIEWER #2

Limitations section: While some limitations are discussed, potential response biases (e.g., social desirability) should be more thoroughly examined.

AUTHORS

We do agree that response biases are always present in self-reported data. This is precisely one reason why our research team has not been doing surveys as frequently as we did in the past. We have become increasingly skeptical about such psychometric methods. However, in the present study, we do not see many opportunities or reasons for the respondents to have employed answering strategies or have been exposed to biases such as social desirability, acquiescence, and others, since they did not express their views about socially shared issues. The only measures they provided referred to their own expected and actual performance (before and after each task). Anyway, we added a note in the limitations section warning that we had not considered such biases as a possible intervening factor, as follows:

“Still concerning the individuals’ traits, we also did not consider if some participants had a significant inclination towards response biases such as social desirability (Fisher, 1993) and acquiescence (Watson, 1992). While some may see this as a limitation*, we had not considered such biases as intervening factors because we had not asked respondents to express their views about socially shared issues.

* We thank a reviewer for this remark.”

Mentioned sources:

Fisher, R. J. (1993). Social desirability bias and the validity of indirect questioning. Journal of Consumer Research, 20(2), 303-315, https://doi.org/10.1086/209351

Watson, D. (1992). Correcting for acquiescent response bias in the absence of a balanced scale: An application to class consciousness. Sociological Methods & Research, 21(1), 52-88, https://doi.org/10.1177/0049124192021001003

---

REVIEWER #2

The study provides an interesting perspective on feedback and self-efficacy, but improvements in research design, statistical analysis, and methodological transparency are necessary. Addressing these concerns will enhance the paper’s contribution to the field. Hope you will find this comments valuable to improve your research article.

AUTHORS

We truly thank the Reviewer for granting us the opportunity to revise our study by taking into consideration numerous insightful comments and critiques.

---

FINAL COMMENTS:

We believe that the revised manuscript offers the due answers to the Reviewers’ concerns as well as an improved reading experience to the prospective reader of Behavioral Sciences. We tried to satisfactorily address each review comment, and we also implemented a few additional improvements by ourselves.

We thank again the Reviewer for the opportunity granted to us to revise our study and the manuscript, and we remain open to discuss further issues if needed.

Sincerely,

The authors

Round 2

Reviewer 1 Report

Comments and Suggestions for Authors

First, I acknowledge that the timeframe for the revision was quite tight, especially given the number of comments and concerns that needed to be addressed. I appreciate that the authors incorporated key topics such as feedforward, which I had suggested, and that they expanded the methodological sections in an effort to address my concerns. Additionally, I recognize the improvements made in the discussion, where they added several pieces of information that were necessary.

However, I still have some concerns. In my initial comment, I noted that the structure of the literature review could be more organized and should have a clearer objective. The authors responded that this comment surprised them and suggested that differences in academic genres across disciplines might explain my perspective, particularly when comparing their field to the behavioral sciences. They explained that they were unable to change the organization of the literature review. While I understand part of their reasoning, I still believe that the objective could be more clearly articulated and better substantiated. I would encourage them to refine the first section further—not just through textual changes but also by improving its structural coherence. Regardless of the academic genre, a well-structured progression toward the research goal is essential for any scholarly article.

Regarding my remark about students sharing information with one another, I appreciate that this point has now been addressed in the discussion section. However, I believe the authors may underestimate the extent to which students communicate with each other. Therefore, the statement that it is "very unlikely that they would share information" seems somewhat risky.

I also still believe that the discussion section concerning the student who left should be revised. My original concern was with the following passage:

"Therefore, this may explain that student’s behavior as well as why other students had difficulty adjusting their CSE levels. Since that student was a woman, another possible explanation, according to a review on self-perceptions by Chevrier et al. (2020), relates to female college students being more self-critical about their academic abilities and more affected by self-esteem issues that address their intellectual abilities, scholastic competence, and expectations about college life."

This explanation seems somewhat reductive—it is based on a single case and does not appear entirely relevant to the broader scope of the study, especially since gender is elsewhere stated as not being a relevant variable. At the very least, this statement should not contradict the earlier assertion that gender is not a determining factor.

Finally, the practical implications could be articulated more specifically. As they stand, they remain somewhat vague regarding the concrete contributions of the study. The utility of the final implication—the tool—also remains unclear. This was a point I raised previously, and I still find it lacking. Perhaps, if the study’s objective is clarified in the introduction, the broader significance of the research and its value for both academic and practical applications will become more evident.

Author Response

Dear Reviewer:

We write this revision letter regarding our manuscript code “behavsci-3473350” (titled “Computer Self-efficacy and Reactions to Feedback: Reopening the Debate in an Interpretive Experiment with Overconfident Students”). We are pleased to know that our previous revision attracted positive impressions by the Reviewer and that there are only a few remaining issues to be solved.

We have read the paper in full again and paid particular attention to addressing the Reviewer's comments both in the revised paper and in this response letter. While the timeframe granted for this revision was even tighter (only three days), we did our best to meet the requests.

COMMENT #1

First, I acknowledge that the timeframe for the revision was quite tight, especially given the number of comments and concerns that needed to be addressed. I appreciate that the authors incorporated key topics such as feedforward, which I had suggested, and that they expanded the methodological sections in an effort to address my concerns. Additionally, I recognize the improvements made in the discussion, where they added several pieces of information that were necessary.

AUTHORS

Thank you indeed for acknowledging the effort we have dedicated to the previous revision.

---

COMMENT #2

However, I still have some concerns. In my initial comment, I noted that the structure of the literature review could be more organized and should have a clearer objective. The authors responded that this comment surprised them and suggested that differences in academic genres across disciplines might explain my perspective, particularly when comparing their field to the behavioral sciences. They explained that they were unable to change the organization of the literature review. While I understand part of their reasoning, I still believe that the objective could be more clearly articulated and better substantiated. I would encourage them to refine the first section further—not just through textual changes but also by improving its structural coherence. Regardless of the academic genre, a well-structured progression toward the research goal is essential for any scholarly article.

AUTHORS

We thank the Reviewer for pushing us again to improve the literature review section. We are not sure if now we meet the Reviewer’s expectation, but we found the opportunity to explain to the reader how the three sub-sections of the literature review connect with the introductory first paragraph of the literature review. That paragraph briefly addressed technology use in both the traditional views and the contemporary perspective of technology use effectiveness, but failed to connect with the following sections of the literature review about self-efficacy, task performance, feedback, and learning. Therefore, now we include multiple notes in the first two paragraphs of the Literature Review section to explain why we review the literature on those four concepts, as follows:

(…) We thus resort to the works of Burton-Jones and Grange (2013) and Porto-Bellini (2018) on technology use effectiveness to discuss the presence of individuals in the educational and professional settings who may not hold accurate views about themselves in terms of the actual competencies needed to perform a task on the computer. The use effectiveness perspective extends traditional views of technology use by focusing on how mindful one is about his or her technology-related motivational attitudes, theoretical abilities, and practical skills. Moreover, effectiveness is a relativistic measure of the level in which someone achieves his or her arbitrarily defined technology use purposes, i.e., each stakeholder in a given technology use situation will hold a particular view of effectiveness in the light of his or her personal perspectives about what effectiveness is (what he or she wishes to do) in that particular situation (Porto-Bellini, 2018).

In the next sections, we review the seminal works and research opportunities on the key concepts of our study (self-efficacy beliefs, task performance, feedback, and learning) and connect them with the technology use effectiveness rationale. Self-efficacy beliefs are instrumental for one to achieve effectiveness in technology use (an individually defined expectation about task performance). Feedback, in its turn, is an opportunity to demonstrate that different perspectives may exist about task performance and that the stakeholders must negotiate a shared expectation as per the demands of the situation. For instance, students and instructors may hold different views about acceptable student performance in the classroom. While all views are acceptable as per the concept of technology use effectiveness (Porto-Bellini, 2018), the stakeholders must share a common expectation if a larger stakeholder is present (in the case of classroom activities, the larger stakeholder is the school’s evaluation system). Finally, learning is a natural process through which people adapt their views about the world and about themselves in order to meet those shared expectations.

---

COMMENT #3

Regarding my remark about students sharing information with one another, I appreciate that this point has now been addressed in the discussion section. However, I believe the authors may underestimate the extent to which students communicate with each other. Therefore, the statement that it is "very unlikely that they would share information" seems somewhat risky.

AUTHORS

We do agree with the Reviewer that we could have anticipated this issue, as there is a possibility that students communicate about a course’s activities. Now we have adjusted some words about this issue in the Limitations section.

The Reviewer’s comment made us think extensively on why this may be an issue for our readers and was not for us during the design of the experiments. We realized that the Reviewer addresses something that has become consensual among our peers in Brazil (where we teach) and possibly also in the USA (where one of the authors worked as well). In both contexts, two issues contribute for our perspective on campus dynamics. First, in the last two decades, we have witnessed a clear decline in students’ involvement with the university environment, such as revealed in direct conversations with students and by analyzing the increasing dropout rates. In direct conversations with students, we realize, for instance, how disconnected they are from the on-campus interactions and opportunities. One evidence of such a reality is that we are not concerned anymore in replicating past exams across classrooms, as we are convinced that past students will not share their memories with the new students. This occurs because they simply do not interact or do not have an interest in discussing their classroom experiences, and the grades reveal just that: the distribution of high and low grades remain the same over the semesters in a same course. And a second issue contributing to our perspective is that the dropout rates impact the formation of each new classroom, in that it is uncommon to have the same students over again across courses. Anyway, such things may be specific to the universities where we have taught.

---

COMMENT #4

I also still believe that the discussion section concerning the student who left should be revised. My original concern was with the following passage:

"Therefore, this may explain that student’s behavior as well as why other students had difficulty adjusting their CSE levels. Since that student was a woman, another possible explanation, according to a review on self-perceptions by Chevrier et al. (2020), relates to female college students being more self-critical about their academic abilities and more affected by self-esteem issues that address their intellectual abilities, scholastic competence, and expectations about college life."

 This explanation seems somewhat reductive—it is based on a single case and does not appear entirely relevant to the broader scope of the study, especially since gender is elsewhere stated as not being a relevant variable. At the very least, this statement should not contradict the earlier assertion that gender is not a determining factor.

AUTHORS

While we do not agree in full that the explanation is irrelevant or reductive, we decided to exclude the last half of that paragraph, including the conjecture on the role of gender in the specific case of that student. Our partial disagreement is due to our (hopefully correct) procedure of using theory to explain a single case, not the opposite (using a single case to develop theory). Also, we did not exclude the possibility of genders playing a role in studies like ours. What we have said about genders in the Methods section was:

(...) For instance, one may question why gender was not a priority for pairing. The reason is, no interactions were found between gender and intelligence theories on overconfidence in Ehrlinger, Mitchum, and Dweck’s (2016) study on overconfidence with undergraduate students. Broadly, those authors found a lack of scholarly knowledge regarding the demographic aspects of overconfident individuals, rather assuming that views of intelligence (entity/fixed or incremental/malleable) and the locus of attention (difficult or easy tasks) may explain self-assessments of performance and effects on learning. On a note of caution, Biernat and Danaher (2012) found differences in immediate reactions to subjective interpretations of feedback according to gender and race, but their study is not comparable to ours in many ways (e.g., their experimental tasks involved leadership roles, and their focal measurement was the level of importance the participants assigned to those roles); and Narciss et al.’s (2014) study found gender differences in learning performance of stu-dents under tutoring schemes, which is also not the case in our study.

Anyway, as said, we decided to exclude that controversial discussion.

---

COMMENT #5

Finally, the practical implications could be articulated more specifically. As they stand, they remain somewhat vague regarding the concrete contributions of the study. The utility of the final implication—the tool—also remains unclear. This was a point I raised previously, and I still find it lacking. Perhaps, if the study’s objective is clarified in the introduction, the broader significance of the research and its value for both academic and practical applications will become more evident.

AUTHORS

We hope to have improved this issue in a new paragraph about the implications for practice, as follows:

This study has implications also for classroom activities, organizational hiring, training, and team building. In summary, we provide three new scales for CSE (Appendix A), two versions of a scenario-based computer-aided decision task for use in the classroom and in professional training (Appendix B), an experimental design, and numerous insights into the significant numbers of overconfident individuals that may be present in a work group (since we found around one-third of overconfident individuals in our sample by using a very conservative measure). Particularly useful is the use of the tool in Appendix B. When teaching topics like the commoditization of IT (Carr, 2003), the productivity paradox (Anderson, Banker, & Ravindran, 2003) and business competition in the IT industry, students want to see real situations to better understand the logic of the Prisoner’s Dilemma in action. The tool we offer can be used in this sense for a first glimpse at the concepts and how decisions are actually made when a company’s strategy needs to take into account a competitor’s strategy. Another possible use of the tool, along with the CSE instrument in Appendix A, is to promote self-reflection by the very participants of behavioral experiments and identify individual decision-making patterns (Folke et al., 2017). Such convenient experiments may efficiently reveal the presence of overconfident individuals in teamwork, as expectations on those individuals, and by them, may not correspond to reality and thus result in frustration, opposition, and discontinuance of work (such as what happened with one overconfident student who abandoned our experiment).

New sources:

Anderson, M. C., Banker, R. D., & Ravindran, S. (2003). The new productivity paradox. Communications of the ACM, 46(3), 91-94, https://doi.org/10.1145/636772.636776  

Carr, N. G. (2003). IT doesn’t matter. Harvard Business Review, May, 5-12.

---

FINAL COMMENTS:

We believe that the revised manuscript offers the due answers to the Reviewer’s concerns as well as an improved reading experience to the prospective reader of Behavioral Sciences.

We thank again the Editor and the Reviewers for the opportunity to revise our manuscript, and we remain open to discuss further issues if needed.

Sincerely,

The authors

Reviewer 2 Report

Comments and Suggestions for Authors

You have now addressed all the posed issues and commented them in separate file and also included them concisely into the article. Therefore I now find this article as appropriate for publishing and congrats the authors.

Author Response

We thank the Reviewer for helping us to improve our work and acknowledging our efforts in the revision.